# GENIUS: Generative Fluid Intelligence Evaluation Suite

## Abstract

Unified Multimodal Models (UMMs) have shown remarkable progress in visual generation. Yet, existing benchmarks predominantly assess *Crystallized Intelligence*, which relies on recalling accumulated knowledge and learned schemas. This focus overlooks *Generative Fluid Intelligence (GFI)*: the capacity to induce patterns, reason through constraints, and adapt to novel scenarios on the fly. To rigorously assess this capability, we introduce **GENIUS** (**GEN**erative Fluid **I**ntelligence Eval**U**ation **S**uite). We formalize *GFI* as a synthesis of three primitives. These include *Inducing Implicit Patterns* (e.g., inferring personalized visual preferences), *Executing Ad-hoc Constraints* (e.g., visualizing abstract metaphors), and *Adapting to Contextual Knowledge* (e.g., simulating counter-intuitive physics). Collectively, these primitives challenge models to solve problems grounded entirely in the immediate context. Our systematic evaluation of 12 representative models reveals significant performance deficits in these tasks. Crucially, our diagnostic analysis disentangles these failure modes. It demonstrates that deficits stem from limited context comprehension rather than insufficient intrinsic generative capability. To bridge this gap, we propose a training-free attention intervention strategy. Ultimately, **GENIUS** establishes a rigorous standard for *GFI*, guiding the field beyond knowledge utilization toward dynamic, general-purpose reasoning.

## 1. Introduction

Unified Multimodal Models (UMMs) have witnessed remarkable progress recently (Team, 2024; Chen et al., 2025; Xie et al., 2024), delivering impressive results across diverse tasks (An et al., 2025; Li et al., 2025a; Jiang et al., 2025).

[1]Anonymous Institution, Anonymous City, Anonymous Region, Anonymous Country. Correspondence to: Anonymous Author <anon.email@domain.com>.

Preliminary work. Under review by the International Conference on Machine Learning (ICML). Do not distribute.

Benefiting from the fusion of understanding, UMMs are capable of processing complex, interleaved contexts and exhibiting extensive world knowledge to reshape the generative paradigm. Consequently, they are widely regarded as a milestone on the path toward Artificial General Intelligence (AGI). However, this rapid advancement invites a natural question: *How far are current UMMs from achieving true general intelligence regrading visual generation?*

To investigate this problem, drawing upon existing literature (Cattell, 1963; Schipolowski et al., 2014; Kent, 2017), we deconstruct General Intelligence in visual generation into two primary components: *Crystallized Intelligence (CI)* and *Fluid Intelligence (FI)*. Current development and evaluation focus of UMMs mainly targets *CI* (i.e., the capacity for memorization and retrieval of pre-trained knowledge). For instance, a model's ability to generate a flawless "cat" often stems from exposure to billions of instances during training, followed by probabilistic reproduction during inference. However, this trend has severely masked a critical but long-ignoring deficiency concerning *FI* of visual generation skills, termed *Generative Fluid Intelligence (GFI)*, (i.e., the ability to perform inducing, reasoning and ad-hoc adaptation in novel scenarios). As shown in Fig. 1, the "Simple Constraint" task requires the model to identify ad-hoc rules (e.g., abstract symbol denotes "rain") and apply them to the visual output, instead of just retrieving static concepts.

Despite its critical importance, research along this direction remains limited (shown in Tab. 1):

- **First, a formal definition is absent.** This theoretical void impedes the foundational guidance, which is necessary for steering UMMs toward general intelligence.
- **Second, benchmarks are inadequate.** Current evaluations predominantly assess model memorization and retrieval, failing to disentangle static knowledge to probe the true bounds of general intelligence.
- **Third, systematic analyses are lacking.** The lack of investigations into the failure modes leaves critical questions of *why models fail* and *how to improve* unanswered.

To bridge these gaps, we introduce **GENIUS** (**GEN**erative Fluid **I**ntelligence Eval**U**ation **S**uite), the first framework dedicated to the systematic evaluation of GFI. Drawing from the Cattell-Horn-Carroll (CHC) theory (Schneider & McGrew, 2012), we distill three core primitives of *FI*: (I) *In-*

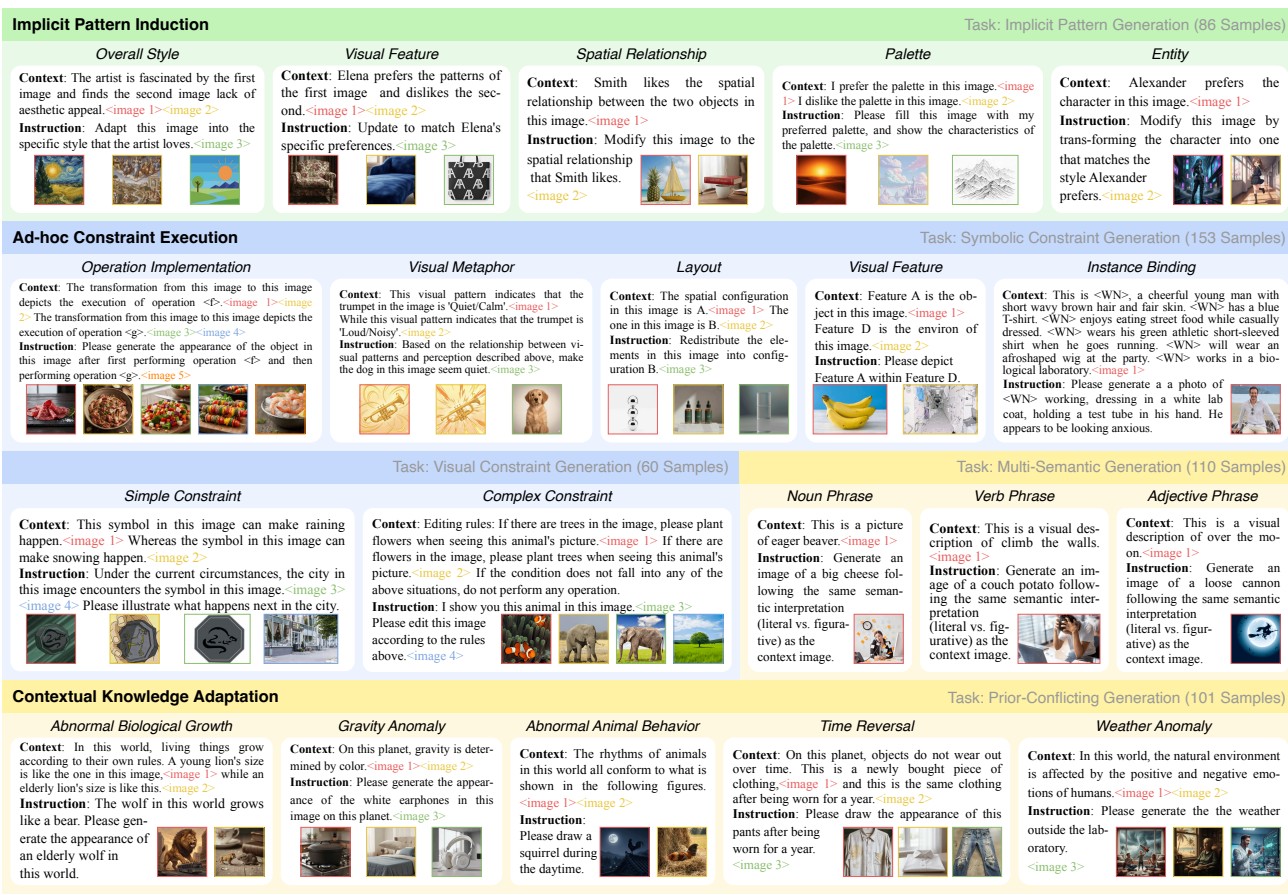

Figure 1. **An overview of GENIUS benchmark.** It is hierarchically structured into three dimensions, five tasks, and diverse sub-tasks.

*ductive Inference*, (II) *Abstract Dynamic Reasoning* and (III) *Adaptive Inhibition*. We materialize these theoretical concepts into three corresponding dimensions within **GENIUS**. To ensure a fine-grained assessment, we further construct five novel and well-designed tasks that specify concrete capabilities within each dimension. For each task, we employ a hybrid evaluation comprising three metrics: (I) *Rule Compliance*, which challenges the model's precision in following ad-hoc rules; (II) *Visual Consistency*, which assesses the stability of generated attributes under logical constraints; and (III) *Aesthetic Quality*, which demands that the model maintains fundamental aesthetic standards. Through manual curation, our suite features tasks well designed by multimodal experts. Unlike traditional benchmarks that prioritize static world knowledge, generation quality or safety, we ensure that every sample presents a dynamic and novel rule, strictly decoupling static knowledge to offer a pure quantification of the model's *GFI* capabilities.

With **GENIUS**, we systematically evaluate 12 representative open-source and proprietary models. To ensure evaluation robustness, we provide manually annotated hints for each test case, which have undergone at least three rounds

of cross-validation to support unbiased hybrid evaluation. Overall, our results reveal clear gaps between current state-of-the-art (SOTA) models and general intelligence. Surprisingly, pre-planning and post-reflection yield marginal gains. These findings expose under-explored deficiencies in current generative models, highlighting the urgent need to advance fluid intelligence in the next generation of UMMs.

Building on these findings, we move beyond evaluation to investigate the underlying mechanisms of failure. Taking Bagel (Deng et al., 2025) as an example, by visualizing the attention distribution of it, it surprisingly shows irregular noise and spikes across the multimodal context. This indicates that the model struggles to accurately focus on critical new rules in the context. Inspired by the theoretical perspective of *In-Context Learning (ICL) as Implicit Fine-Tuning* (Dherin et al., 2025), we demonstrate that ICL process is mathematically equivalent to update specific model parameters while generation. Then, we offer a possible view: the imbalanced attention distribution results in insufficient guidance during implicit gradient descent, which causes the gradient direction vague or stochastic, failing to overcome the inertia of pre-trained priors. Based on this, we design

*Table 1.* **Comparison of representative benchmarks.** * denotes understanding tasks. ✔ indicates partial satisfaction (e.g., For Manual Curated/Annotation, it implies a combination of human curation and automatic methods). GENIUS pioneers Fluid Intelligence evaluation, featuring multi-image input, multimodal interleaved context, hybrid metrics, and pure manually curated and annotated testcases.

| Benchmark | # Samples | Multi-Image Input | Fluid Intelligence | Multimodal Interleaved Context | Task Dimension | | | Hybrid Evaluation | Manual Curated Annotation |
| --- | --- | --- | --- | --- | --- | --- | --- | --- | --- |
| | | | | | Implicit Pattern Induction | Explicit Constraint Execution | Contextual Knowledge Adaptation | | |
| GenEval (Ghosh et al., 2023) | 2.2k | ✗ | ✗ | ✗ | ✗ | ✗ | ✗ | ✗ | ✗ |
| WISE (Niu et al., 2025) | 1.0k | ✗ | ✗ | ✗ | ✗ | ✗ | ✗ | ✗ | ✔ |
| RISE (Zhao et al., 2025) | 360 | ✗ | ✗ | ✗ | ✗ | ✗ | ✗ | ✓ | ✓ |
| DPG-Bench (Hu et al., 2024) | 1.0k | ✗ | ✗ | ✗ | ✗ | ✗ | ✗ | ✗ | ✗ |
| DreamBooth (Ruiz et al., 2023) | 3.0k | ✗ | ✗ | ✗ | ✗ | ✗ | ✗ | ✗ | ✓ |
| UnifyBench (An et al., 2025) | 0.1k | ✗ | ✗ | ✗ | ✗ | ✔ | ✗ | ✗ | ✓ |
| Tiif-Bench (Wei et al., 2025) | 5.0k | ✗ | ✗ | ✗ | ✗ | ✔ | ✗ | ✗ | ✔ |
| OpenING* (Zhou et al., 2025) | 5.4k | ✗ | ✗ | ✗ | ✗ | ✗ | ✗ | ✗ | ✔ |
| MME-Unify* (Xie et al., 2025b) | 4.1k | ✓ | ✗ | ✗ | ✗ | ✗ | ✗ | ✗ | ✔ |
| UniEval* (Li et al., 2025b) | 4.2k | ✗ | ✗ | ✗ | ✗ | ✗ | ✗ | ✗ | ✗ |
| MME-Unify* (Xie et al., 2025b) | 4.1k | ✓ | ✗ | ✗ | ✗ | ✗ | ✗ | ✗ | ✔ |
| RealUnify* (Shi et al., 2025) | 1.0k | ✓ | ✗ | ✗ | ✗ | ✗ | ✗ | ✗ | ✓ |
| ROVER (Liang et al., 2025) | 1.3k | ✓ | ✗ | ✗ | ✗ | ✗ | ✗ | ✓ | ✓ |
| WEAVE (Chow et al., 2025) | 0.1k | ✓ | ✗ | ✔ | ✗ | ✗ | ✗ | ✓ | ✓ |
| **GENIUS (Ours)** | 0.5k | ✓ | ✓ | ✓ | ✓ | ✓ | ✓ | ✓ | ✓ |

a training-free mechanism as a strong baseline. The results show consistent performance gains across all tasks, which not only validates the effectiveness of our method but also corroborates the rationality of our theoretical framework.

In summary, our core contributions are as follows:

- We formally define *Generative Fluid Intelligence (GFI)*, filling a theoretical void to provide foundational guidance for steering UMMs toward general intelligence.
- We introduce **GENIUS**, the first benchmark designed to purely quantify *GFI*. It features 510 expert-curated samples spanning three dimensions and five tasks, supported by a robust hybrid evaluation protocol.
- We systematically evaluate 12 representative open-source and proprietary models. Results reveal significant deficits in SOTA models, underscoring the clear performance gap.
- We trace *GFI* failures via theoretical and empirical analysis. Then, we propose a training-free strategy that boosts performance, effectively activating the model's *GFI*.

## 2. GENIUS

### 2.1. Benchmark Overview

Grounded in the Cattell-Horn-Carroll (CHC) theory (Schneider & McGrew, 2012), General Intelligence is defined not merely by *Crystallized Intelligence* but more fundamentally by *Fluid Intelligence*, which is the capacity to induce, reason, and adapt in novel scenarios, independent or even contrary to prior knowledge. Drawing from this, we formally define *Generative Fluid Intelligence (GFI)* for visual generation as the synthesis of three core primitives, where humans effortlessly demonstrate these capabilities through:

- **Induce Implicit Pattern** (Inductive Inference): Distilling implicit patterns and intrinsic attributes from observations.
- **Execute Ad-hoc Constraints** (Abstract Dynamic Rea-

soning): Executing logical reasoning within the bound of ad-hoc defined visual or symbolic constraints.
- **Adapt based on Contextual Knowledge** (Adaptive Inhibition): Adjusting based on contextual cues, even when necessitating a deviation from established common sense.

However, these capabilities still present significant challenges for current UMMs. To objectively assess model performance in these areas and pinpoint limitations, we introduce **GENIUS** (**GEN**erative Fluid **I**ntelligence Eval**U**ation **S**uite), the first benchmark dedicated to evaluate *Generative Fluid Intelligence*. As illustrated in Fig. 5, the suite comprises a total of 510 expert-curated samples spanning 20 diverse sub-tasks, structurally distributed across three core dimensions: 86 for Implicit Pattern Induction, 213 for Ad-hoc Constraint Execution, and 211 for Contextual Knowledge Adaptation. Unlike previous benchmarks that prioritize static world knowledge, generation quality or safety, **GENIUS** is constructed to strictly exclude prior knowledge. It systematically evaluates models across (I) Inductive Inference, (II) Abstract Ad-hoc Reasoning, and (III) Adaptive Inhibition, as aforementioned, purely quantifying their aptitude for solving novel problems.

### 2.2. Benchmark Construction

For each category of **GENIUS**, we curate a diverse set of high-quality, expert-designed test cases. Each instance comprises a multi-modal interleaved context and removal of any single modality from context makes the task unsolvable.

**Implicit Pattern Induction:** This dimension mainly contains a novel task: *Implicit Pattern Generation*, which assesses the capacity to deduce unstated visual preferences from context and apply them for generation. As shown in Fig. 1, the interleaved input presents images of varying styles alongside specific user preferences. During testing, the model is required to induce the target stylistic pattern

based on these preferences and manifest it in the generated output. This task necessitates the integration of both modalities: relying solely on images would cause the model to blindly conflate visual features, while relying solely on text would leave the stylistic preference undefined, causing the model to collapse into its pre-trained distribution.

**Ad-hoc Constraint Execution:** A significant ability of *FI* is to perform dynamic reasoning under newly defined, ad-hoc rules. To systematically assess this, we construct two complementary tasks: (I) *Visual Constraint Generation* (II) *Symbolic Constraint Generation*, where novel meanings are assigned to symbols or images within the context. Crucially, we deliberately select elements that are devoid of pre-existing semantic associations, such as an image patch (e.g., "defining a plain blue square as an operation to remove the specifc object") or a mathematical symbol (e.g., "defining a function $f$ as an instruction to make the object melt"). This design reflects the model's capability to solve novel problems by reasoning abstractly. The absence of either visual or textual modalities would fundamentally compromise the establishment of these ad-hoc rules, making it impossible to link the element to its new definition.

**Contextual Knowledge Adaptation:** A model possessing *FI* must exhibit flexible adaptation rather than rigid adherence to pretrained knowledge. We introduce two novel tasks to evaluate this: (I) *Prior-Conflicting Generation:* The model must reason based on newly defined common sense presented in the context (e.g., "object weight is determined by color"), even if it contradicts established facts. (II) *Multi-Semantic Generation:* This task requires the model to discern whether to interpret a concept literally or metaphorically (e.g., distinguishing "a green hand" as a novice vs. green skin) based on the specific multimodal contexts. In both tasks, missing any modality prevents the precise definition of new knowledge or the clarification of semantics, causing the model to fail in dynamic adaptation.

### 2.3. Evaluation Metric

Evaluating the ability of *GFI* remains a challenging task. Recent studies (Gu et al., 2024) have shown that large multimodal models (LMMs) exhibit strong visual reasoning and alignment capabilities, making them suitable as evaluators. Following this, we construct a robust pipeline comprising three orthogonal metrics. We utilize the frontier LMM (Gemini-3-Pro (Google DeepMind, 2025)) as the judge, employing structured prompts (Detailed in the Appendix A.2) and a hybrid evaluation strategy that incorporates manually-curated hint to ensure rigorous quantification. All metrics are assigned a score of 0 (fail), 1 (partial), or 2 (perfect).

**Rule Compliance** This metric measures the model's accuracy in executing strict ad-hoc rules. Since *GFI* tasks involve precise constraints (e.g., specific symbolic, layouts

or palette), relying on the LMM's unguided interpretation is unreliable. We therefore adopt a hybrid protocol that grounds automated evaluation in human-verified truth. For each sample, we provide a manually curated eval-hint serving as the "gold standard". The LMM evaluator strictly compares the output against the specific nouns, adjectives and spatial predicates defined in this eval-hint.

**Visual Consistency** For some *GFI* tasks, the visual identity of original objects must remain unchanged (e.g., specific characters or objects). This metric assesses the model's ability to preserve context during dynamic reasoning. We provide specific hybrid eval-hints for each sample that identify the key visual elements from the reference images. The evaluator verifies the stability of these elements in the generated image, ensuring the model does not hallucinate or discard critical context while following instructions.

**Aesthetic Quality** Generative outputs must maintain physical coherence even when processing novel or conflicting semantic inputs. We employ a specific prompt to evaluate aesthetic quality, focusing on anatomical logic, lighting, and the presence of AI artifacts (e.g., distorted limbs). This metric ensures that the model's adaptation to new rules does not come at the cost of basic visual realism.

## 3. Experiment

We conduct a comprehensive evaluation of 12 representative open-source and proprietary models. The open-source model comprises Qwen-Image-Edit-2511 (Wu et al., 2025b), GLM-Image (Zhipu AI Team, 2026), FLUX.2-dev (Labs, 2025), NextStep-1 (Team et al., 2025), Emu3.5-Image (Cui et al., 2025) and Bagel (Deng et al., 2025). The proprietary category includes leading commercial models: Nano Banana (Google, 2025a) and its Pro variant (Google, 2025b), SeeDream series (4.0 & 4.5) (Seedream et al., 2025) and GPT-Image (OpenAI Team, 2025).

As outlined in Sec. 2.3, we employ Gemini-3-Pro (Google DeepMind, 2025) as the evaluator. To mitigate stochastic variance and ensure robustness, we report the final scores as the average of three independent runs for each sample. The quantitative results are shown in Tab. 3. Given the diversity of multimodal input formats, we adopt interleaved inputs for models capable of processing them, while utilizing a decoupled format for those that do not. Further ablation studies concerning interleaved formats are in the Appendix D.1.

### 3.1. Main Results

***Generative Fluid Intelligence (GFI)* remains a significant bottleneck for current models.** Our results reveal a stark reality: even the state-of-the-art proprietary model, Nano Banana Pro, achieves an overall score of only 57.19, falling short of a passing grade. Meanwhile, representative open-

*Table 2.* **Main Results.** We evaluate models across dimensions. The **Overall** column represents weighted score across all tasks, calculated using a ratio of RC:VC:AQ = 6:3.5:0.5. Ours is implemented on Bagel. The best and second best performances are highlighted.

| Method | Interleaved | Overall | Implicit Pattern Induction | | | Ad-hoc Constraint Execution | | | | | | Contextual Knowledge Adaptation | | | | | |
|---|---|---|---|---|---|---|---|---|---|---|---|---|---|---|---|---|---|
| | | | Implicit Pattern | | | Symbolic Constraint | | | Visual Constraint | | | Prior-Conflicting | | | Multi-Semantic | | |
| | | | RC | VC | AQ | RC | VC | AQ | RC | VC | AQ | RC | VC | AQ | RC | VC | AQ |
| *Proprietary Models* | | | | | | | | | | | | | | | | | |
| Nano Banana Pro | ✓ | 57.19 | 66.86 | 44.59 | 96.51 | 71.38 | 50.00 | 92.11 | 76.67 | 66.67 | 96.67 | 52.97 | 41.38 | 90.59 | 35.45 | - | 95.00 |
| Nano Banana | ✓ | 50.66 | 56.47 | 39.04 | 94.12 | 60.46 | 51.91 | 90.20 | 68.33 | 79.17 | 93.33 | 35.50 | 39.47 | 91.00 | 30.28 | - | 93.12 |
| GPT-Image | ✗ | 47.15 | 58.14 | 41.92 | 93.60 | 58.82 | 32.82 | 93.79 | 49.17 | 62.50 | 92.50 | 43.50 | 33.33 | 90.00 | 28.64 | - | 85.45 |
| SeeDream 4.0 | ✗ | 21.26 | 12.05 | 0.70 | 96.39 | 21.57 | 3.44 | 84.64 | 40.00 | 4.17 | 76.67 | 30.69 | 10.34 | 82.67 | 30.73 | - | 80.00 |
| SeeDream 4.5 | ✗ | 52.84 | 70.00 | 59.59 | 97.06 | 62.91 | 41.09 | 94.37 | 58.33 | 62.50 | 86.67 | 40.10 | 41.38 | 92.57 | 35.00 | - | 86.82 |
| *Open-Source Models* | | | | | | | | | | | | | | | | | |
| Qwen-Image | ✗ | 30.58 | 36.18 | 27.69 | 71.05 | 36.18 | 27.69 | 71.05 | 26.67 | 45.83 | 55.83 | 27.72 | 20.69 | 71.78 | 25.91 | - | 69.55 |
| GLM-Image | ✗ | 24.71 | 32.94 | 19.86 | 93.53 | 22.37 | 21.15 | 87.50 | 27.50 | 12.50 | 70.83 | 20.30 | 15.52 | 71.29 | 17.73 | - | 70.91 |
| FLUX.2-dev | ✗ | 34.39 | 34.30 | 27.70 | 88.95 | 35.76 | 31.01 | 87.09 | 39.17 | 50.00 | 59.17 | 25.25 | 30.17 | 84.16 | 29.82 | - | 79.82 |
| NextStep-1 | ✗ | 10.44 | 10.74 | 0.40 | 25.12 | 11.33 | 2.54 | 21.67 | 21.50 | 4.20 | 29.17 | 15.49 | 7.55 | 28.71 | 12.80 | - | 20.28 |
| Emu3.5-Image | ✗ | 36.67 | 41.86 | 35.81 | 83.72 | 34.97 | 39.31 | 86.93 | 24.17 | 29.17 | 42.50 | 26.24 | 37.93 | 82.18 | 32.87 | - | 75.46 |
| Omini-Gen2 | ✗ | 27.87 | 29.07 | 26.35 | 76.16 | 25.33 | 30.38 | 77.96 | 11.67 | 41.67 | 52.50 | 23.76 | 34.48 | 69.80 | 19.27 | - | 63.76 |
| Bagel | ✓ | 26.74 | 26.74 | 27.03 | 84.30 | 29.61 | 16.03 | 76.32 | 22.50 | 12.50 | 49.17 | 22.28 | 17.24 | 74.75 | 33.49 | - | 53.67 |
| Ours | ✓ | 38.84 | 47.34 | 48.51 | 71.88 | 40.71 | 30.81 | 70.02 | 32.77 | 38.56 | 50.51 | 28.78 | 40.87 | 62.96 | 34.81 | - | 56.81 |

source models like Bagel fall significantly behind, scoring a mere 26.74. These tasks demand ad-hoc reasoning and dynamic adaptation to novel rules, which are less directly grounded by the models' pre-trained parametric knowledge. Together, these quantitative deficits suggest that while current UMMs have acquired robust capabilities for crystallized reproduction, they remain fundamentally distant from the fluid adaptability required for general-purpose generation.

**Current models fail to effectively arbitrate the conflict between pre-trained priors and the given context.** As shown in Tab. 2, this deficiency is most pronounced in the *Contextual Knowledge Adaptation* dimension, where performance consistently drops below other task categories. When ad-hoc instructions explicitly contradict world knowledge (e.g., counter-intuitive physical laws or remapped semantics), models exhibit a strong "cognitive inertia", frequently defaulting to their pre-trained priors. This suggests that existing architectures lack a robust mechanism to inhibit intrinsic priors, failing to dynamically adapt to the context.

**Aesthetic fidelity masks deep logical deficiencies.** Our hybrid metric analysis uncovers a pervasive "illusion of competence": models consistently maintain high *Aesthetic Quality* scores, yet their performance on *Rule Compliance* lags substantially behind. This discrepancy suggests that previous model optimization has disproportionately focused on surface-level visual plausibility at the expense of deep context interpretation and logical adherence. By exposing this, **GENIUS** signals a necessary paradigm shift for next generation of models: moving beyond merely generating "beautiful" pixels to achieving profound context comprehension and ensuring logically correct visual synthesis.

### 3.2. Discussion and Analysis

**Pre-planing and post-reflection yield marginal gains.** We investigated various inference-time enhancement strategies to potentially mitigate performance deficits. Taking Nano Banana Pro and Bagel as examples, we implemented pre-planning (activating reasoning mode) and post-reflection (an iterative process where initial generations are evaluated and re-fed as context for refinement). However, as illustrated in Fig. 2(a), empirical results across both Nano Banana Pro and Bagel indicate that these strategies yield only marginal gains. This suggests that current architectures struggle to effectively leverage explicit reasoning for generation.

**Context comprehension is the key to solve *GFI* problems.** To isolate the source of failure, we introduced human-curated hints to guide the generation process. Specifically, we employed a progressive intervention strategy: initially utilizing text-only hints, and subsequently constructing multimodal hints to ensure information completeness, thereby explicitly guiding the model's generation. The results are illustrated in Fig. 2(a). This intervention resulted in substantial performance improvements; however, the degree of improvements varied significantly: Nano Banana Pro exhibited a much more boost compared to Bagel. This observation highlights that accurate context comprehension acts as a critical factor in solving *GFI* tasks. Meanwhile, solving *GFI* problems requires not only accurate context comprehension to decode ad-hoc rules but also robust intrinsic model capabilities to execute them, implying that comprehension aids cannot fully compensate for a weaker base model's generative limitations.

*Figure 2.* **Diagnostic analysis and metric validation.** (a) Performance comparison across different context settings. (b) Analysis of the gap between context comprehension (VQA) and generation capabilities. (c) Correlation analysis validating the LMM-as-a-Judge metric.

**Generative failure primarily stems from an execution gap rather than comprehension deficits.** To investigate the root cause, we reformulated the generative tasks into comprehension-oriented Visual Question Answering (VQA) probes. Specifically, we structured these probes as multiple-choice questions that query the model regarding the expected visual appearance of the target image. We utilized our expert hints for *Rule Compliance* as the ground truth answers, while simultaneously constructing three distractors for each sample to facilitate evaluation. The results are shown in Fig. 2(b). Empirical results reveal a significant disparity: models frequently demonstrate an accurate understanding of the context's intent but fail to translate this into compliant visual outputs. This suggests that the model's current cognitive processing of the context, while sufficient for discriminative understanding tasks, lacks the granularity required for generative reconstruction. We hypothesize this stems from two factors: first, the high information density of interleaved contexts, where fine-grained visual nuances (e.g., specific textures) are difficult to fully capture and articulate through limited modalities; and second, a structural inefficiency in current UMM architectures, where rich semantic understanding from the encoder is not effectively propagated to the generative decoder, resulting in a "know-but-cannot-draw" phenomenon. We further discuss how to enhance this critical contextual comprehension in Sec. 4.1.

### 3.3. Validity of LMM-as-a-Judge

To verify the reliability of using LMMs as a judge, we conducted an analysis of the correlation between LMM-based automated scoring and human expert judgment. We performed a study by randomly and uniformly sampling 100 output images across various dimensions from two representative models: Nano Banana Pro and Bagel. Five human experts were invited to independently rate these samples, adhering to the same metrics used by the LMM evaluator to compare the consistency between human and LMM scoring.

As shown in Fig. 2(c), the Pearson correlation between human expert ratings and LMM-based scores demonstrates a high degree of alignment. Our analysis reveals exceptionally strong global consistency across all samples: the Pearson correlation coefficient ($r$) reaches 0.9630 for NanoBanana Pro and 0.9659 for Bagel. Such high linear correlation indicates that the LMM evaluator accurately captures the underlying logic of human judgment in image generation tasks. Furthermore, dimension-specific analysis shows that the Mean Absolute Error (MAE) remains consistently low across multiple metrics, ranging from 0.06 to 0.11. Relative to the 0–2 scoring scale, these errors are quite small, further validating the robustness of the evaluation framework across different models and task dimensions. In conclusion, the LMM-as-a-Judge framework serves as a reliable and effective alternative to human evaluation.

To further ensure the reproducibility and cross-model robustness, we extended our validation to include the open-source Qwen2.5-VL-72B (Bai et al., 2025) as the judge. Empirical results shown in the Appendix C indicate that while Qwen2.5-VL-72B tends to assign systematically lower absolute scores compared to Gemini-3-Pro, suggesting a stricter evaluation criterion. The relative performance trends and model rankings remain identical. This consistency across proprietary and open-source evaluators confirms that the observed performance gaps are intrinsic to the models being tested rather than artifacts of a specific judge, thereby reinforcing the reliability and generalizability of the results.

## 4. A Potential Solution

Evaluation on **GENIUS** reveals a clear gap between current SOTA models and general intelligence. To diagnose the potential causes of this deficit, we conduct a comprehensive analysis from both theoretical and empirical perspectives, focusing on the widely applicable Bagel framework.

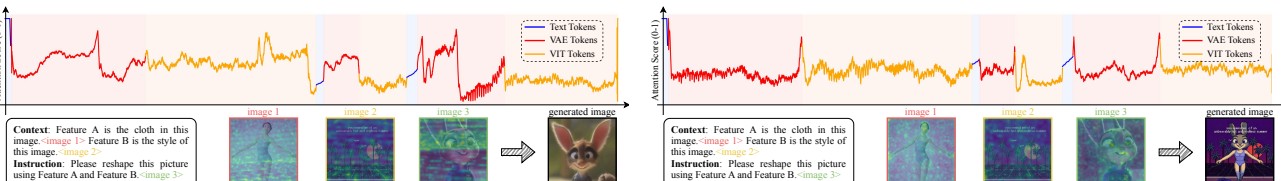

*Figure 3.* **Visualization of attention scores (range [0, 1]).** Left: Existing models. Right: Ours.

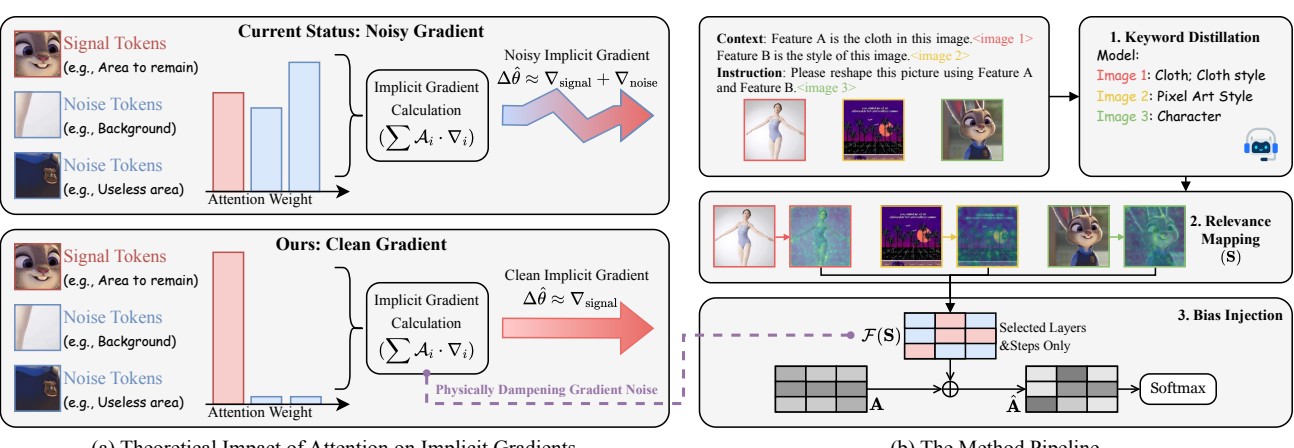

(a) Theoretical Impact of Attention on Implicit Gradients

(b) The Method Pipeline

*Figure 4.* **Method overview.** Guided by the theoretical insight that attention magnitude dictates gradient norms (a), we implement a three-stage pipeline (b) to explicitly suppress noise tokens and rectify the implicit optimization direction.

## 4.1. Experimental Observation

To investigate the underlying mechanism of failure, we visualized the attention distribution over the entire context, using the image tokens generated during the process as the query. Surprisingly, as shown in the left part of Fig. 3, we found the attention distribution is unreasonable: it exhibits irregular noise and stochastic spikes across the multimodal context. This indicates the model struggles to precisely capture pivotal ad-hoc rules from the context. Instead of pinpointing the critical definition, the attention is spread out indiscriminately across the input. As a result, the model fails to extract the specific signal needed for adaptation and simply falls back to its pre-trained priors.

## 4.2. Theoretical Analysis

To explain this phenomenon, we adopt the theoretical perspective of *In-Context Learning (ICL) as Implicit Fine-Tuning* from (Dherin et al., 2025; Ahn et al., 2023; Dai et al., 2023; von Oswald et al., 2023). we perform a derivation on Bagel, which adopts a Mixture-of-Transformer architecture. Since **GENIUS** targets the generative task, we redefine the function $\mathcal{A}(u, g)$ from (Dherin et al., 2025), based on the Bagel model, where $\mathcal{A}$ denotes the network layer component responsible for context processing, $u$ represents the encoding of context and instructions, and $g$ denotes the encoding of intermediate noisy tokens of images. We suppose in the $t$-th step and the $l$-th Decoder blocks we have

$g^{(t,l+1)} = \mathcal{L}^{(l)}_{\text{Up},b}(u^{(l)}, g^{(t,l)})$, where Up is a projection layer in the decoder block, $b$ is the bias of Down layer, and $\mathcal{L}$ represents the $l$-th block's forward propagation. And then we can formalize the relationship between $u$ and the (Up, $b$):

**Theorem 4.1.** *The layer update satisfies following property:*

$$\mathcal{L}_{Up+\Delta Up,\, b+\Delta b}(u', g) = \mathcal{L}_{Up,b}(u, g) \quad (1)$$

*where the bias perturbation is defined as:*

$$\Delta b = \mathcal{A}(u, g) - \mathcal{A}(u', g) \quad (2)$$

*and the upsampling operator perturbation be defined as:*

$$\Delta Up = \frac{Up\,(\delta\mathcal{A})\,\mathcal{N}(\mathcal{A}(u', g))^{\top}}{\|\mathcal{N}(\mathcal{A}(u', g))\|^2}, \mathcal{N}(x) = \frac{x}{RMS(x)} \quad (3)$$

*And the normalized attention difference is given by:*

$$\delta\mathcal{A} = \mathcal{N}(\mathcal{A}(u, g)) - \mathcal{N}(\mathcal{A}(u', g)) \quad (4)$$

According to Thm. 4.1, we can demonstrate that in multimodal generation, the ICL process is mathematically equivalent to updating specific model parameters. This successfully extends the conclusions of (Dherin et al., 2025):

We first formalize the vector notations for clarity: let $u = (u_1, \ldots, u_n)$, and $u^{(i)} = (u_1, \ldots, u_i)$. Then we hope:

$$\mathcal{L}_{\text{Up}_i,b_i}(u^{(i)}, g) = \mathcal{L}_{\text{Up},b}(u, g) \quad \forall i = 1, 2, \ldots, n \quad (5)$$

From this objective, we derive expressions for $\text{Up}_i$ and $b_i$:

$$\text{Up}_i = \text{Up} + \Delta\text{Up}_i = \text{Up} + \frac{\text{Up}\left(\delta\mathcal{A}_i\right)\mathcal{N}(\mathcal{A}(g))^{\top}}{\|\mathcal{N}(\mathcal{A}(g))\|^2} \quad (6)$$

$$b_i = b + \Delta b_i = b + \mathcal{A}(u^{(i)}, g) - \mathcal{A}(g) \quad (7)$$

where the attention difference term $\delta\mathcal{A}_i$ is defined as:

$$\delta\mathcal{A}_i = \mathcal{A}(u^{(i)}, g) - \mathcal{A}(g). \quad (8)$$

Based on the above derivations, we present the key theorem for iterative parameter updates:

**Theorem 4.2.** *For the $(i+1)$-th iteration, Up and b follow the gradient descent update rules below:*

$$\begin{cases} Up_{i+1} = Up_i - h\nabla_{Up}L_i(Up_i), \\ b_{i+1} = b_i - \nabla_b\left(\text{tr}\left(\delta_i^{\top}b_i\right)\right) \end{cases} \quad (9)$$

where $h = 1/\|\mathcal{N}(\mathcal{A}(g))\|^2$ denotes the learning rate. $L_i(\text{Up}) = \text{tr}\left(\Delta_i^{\top}\text{Up}\right)$ is loss function, among which $\Delta_i = \text{Up}\left(\hat{\delta}_i\right)\mathcal{N}(\mathcal{A}(g))^{\top}$, $\hat{\delta}_i = \mathcal{N}(\mathcal{A}(u^{(i)}, g)) - \mathcal{N}(\mathcal{A}(u^{(i+1)}, g))$, and $\delta_i = \mathcal{A}(u^{(i)}, g) - \mathcal{A}(u^{(i+1)}, g)$. Combining empirical observations with theoretical analysis, we offer a hypothesis for the deficit in *GFI*: The imbalanced attention distribution results in a lack of guidance during implicit gradient descent. Consequently, the descent direction becomes stochastic, failing to overcome the pre-trained priors. Full proof of both theorems is in the Appendix G.

### 4.3. Attention Adjustment Mechanism

Guided by Thm. 4.2, we recognize that the magnitude of attention assigned to the context directly dictates the norm of the implicit gradient update. The irregular attention distribution within the context images, as previously observed in Sec. 4.1, implies that irrelevant "noise" tokens currently contribute significant, erroneous gradient components, thereby diverting the optimization trajectory away from the optimal path. To counteract this, we propose a training-free adjustment mechanism to recalibrate the update direction. By explicitly suppressing the attention weights of noise tokens, we mathematically dampen their corresponding gradient norms (i.e., $\|\Delta\text{Up}_{\text{noise}}\| \to 0$), ensuring the implicit fine-tuning is driven solely by critical context signals.

Specifically, we implement this mechanism through a three-stage pipeline shown in Fig. 4(b). First, in the Keyword Distillation phase, leveraging the semantic reasoning capability of Bagel, we prompt the model to distill task-critical visual cues into a set of region-specific keywords $\mathcal{K}$ (the prompt is detailed in the Appendix F.1). Subsequently, during Relevance Mapping, we compute a semantic relevance

map $\mathbf{S}$ by evaluating the alignment between these keywords and the visual context tokens, where $\mathbf{S}$ serves as a proxy for the token's contribution to the effective gradient signal. Finally, via Bias Injection, we inject a spatial bias $\mathcal{F}(\mathbf{S})$ directly into the attention logits:

$$\text{Attention} = \text{softmax}\left(\frac{\mathbf{A} + \lambda \cdot \mathcal{F}(\mathbf{S})}{\sqrt{d}}\right)V \quad (10)$$

This formulation ensures tokens with high relevance are emphasized while noise is suppressed. The detailed mathematical formulation is provided in the Appendix F.2. By rectifying the attention landscape, we re-weight the implicit gradient updates, deterministically steering the optimization trajectory to overcome pre-trained priors.

### 4.4. Experimental Results

As visualized in Fig. 3 (Right), our mechanism successfully rectifies the originally disordered attention landscape into a sharpened distribution with distinct peaks focused on critical tokens. Quantitative results in Tab. 2 further demonstrate consistent performance gains across all dimensions compared to the baseline Bagel (e.g., boosting the Overall score of 12.1%). This validates that deterministically steering the implicit gradient trajectory effectively activates the model's latent *GFI* without requiring parameter updates. Consequently, this mechanism establishes a strong baseline, offering a simple paradigm for improving *GFI* capabilities.

## 5. Conclusion

In this paper, we introduced **GENIUS**, the first benchmark dedicated to systematically quantifying *Generative Fluid Intelligence (GFI)*. By grounding in the Cattell-Horn-Carroll (CHC) theory, we formalized *GFI* into three core dimensions, including *Implicit Pattern Induction*, *Ad-hoc Constraint Execution*, and *Contextual Knowledge Adaptation*, providing a rigorous standard for assessing model capability in novel, reasoning-intensive scenarios. Through systematic evaluation of 12 representative open-source and proprietary models, we reveal a stark reality: even state-of-the-art models like Nano Banana Pro fall short of a passing grade, while open-source models exhibit significant performance deficits. Our analysis exposes a critical "execution gap", where models struggle to arbitrate conflicts between pre-trained priors and ad-hoc context, often prioritizing aesthetic fidelity over logical rule compliance. Furthermore, we partially trace these failures to attention mechanism defects during inference and propose a training-free adjustment strategy that effectively activates latent *GFI* capabilities. We hope that **GENIUS** will serve as a pivotal testbed for future research, guiding the evolution of next-generation models from crystallized memorization toward true general intelligence.

## Impact Statement

This paper presents a benchmark and theoretical framework aimed at advancing the evaluation of *Fluid Intelligence* in generative models. By distinguishing *Generative Fluid Intelligence (GFI)* from standard crystallized knowledge retrieval, our work intends to shift the community focus toward developing systems that possess true adaptability and logic-grounded control. Our contributions align with the goal of creating more robust AI systems by highlighting the "illusion of competence," a phenomenon where aesthetic quality masks logical deficiencies. This focus encourages transparent evaluation and prevents the deployment of models that appear capable but fail in critical, rule-bound scenarios. Furthermore, improved *GFI* capabilities may contribute to versatile creative tools and scientific visualization assistants that can accurately follow complex, ad-hoc instructions without hallucination. We do not foresee any unique negative societal consequences beyond those already recognized in the broader field of generative AI.

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

# A. Benchmark Details

## A.1. Data Statistics

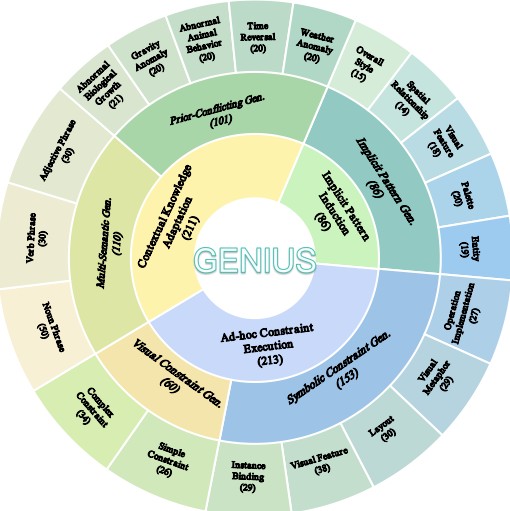

*Figure 5.* **Data composition pie chart. GENIUS** comprises 3 dimensions, 5 tasks, and 20 sub-tasks.

## A.2. Evaluation Prompt

As illustrated in the previously presented prompt templates, we developed a systematic evaluation framework using Large Multimodal Models (LMMs) to assess three key dimensions of generative quality:

**Rule Compliance (RC)**: For each **GENIUS** sample, an audit of textual-visual alignment is conducted. This process rigorously verifies nouns, adjectives, and spatial constraints to ensure 100% compliance with specific modification requests. Details of the prompt template are provided in Fig. 8.

**Visual Consistency (VC)**: For each **GENIUS** sample, *Visual Consistency* may be evaluated multiple times or not at all, depending on how many reference images (objects in the image) need to remain visually consistent. Since we have observed that many open-source models directly copy reference images to cheat (e.g., Bagel, GLM-Image, etc.), a dedicated anti-plagiarism screening is conducted prior to the *Visual Consistency* audit. The LMM first performs a pixel-level identity check; if the target image is found to be an exact pixel-for-pixel duplicate of the reference without any generative modifications, the consistency score is automatically set to 0. Details of the prompt template are provided in Fig. 9.

**Aesthetic Quality (AQ):** Assesses visual logic, rendering clarity, and realism. It rewards commercial-grade outputs while penalizing structural collapses or AI hallucinations. Details of the prompt template are provided in Fig. 10.

# B. Detailed Qualitative Examples and Model Outputs

To provide a more granular view of the **GENIUS** benchmark, we present comprehensive qualitative examples for each sub-task. For every data sample, we showcase a complete data instance that includes: (1) the full input content (comprising both context and instruction); (2) the specific evaluation hints utilized for assessing *Rule Compliance (RC)* and *Visual Consistency (VC)*; and (3) the corresponding generated outputs from six representative models: Nano Banana Pro (Google, 2025b), Nano Banana (Google, 2025a), SeeDream4.5 (Seedream et al., 2025), FLUX.2-dev (Labs, 2025), Bagel (Deng et al., 2025) and ours. These detailed comparisons, which highlight the capabilities and failure modes of different architectures, are illustrated in Fig. 11 and Fig. 12.

# C. Evaluation using Qwen2.5-VL-72B as Judge

To further validate the robustness of our evaluation framework, we employed Qwen2.5-VL-72B (Bai et al., 2025) as the judge model to assess **GENIUS** benchmark. The results are summarized in Tab. 3.

*Table 3.* **Benchmark Results by Qwen2.5-VL-72B.** The **Overall** column represents the weighted score across all tasks, calculated using a metric ratio of RC:VC:AQ = 6:3.5:0.5. The best and second best performances are highlighted.

| Method | Interleaved | Overall | Implicit Pattern Induction | | | Ad-hoc Constraint Execution | | | | | | Contextual Knowledge Adaptation | | | | | |
| | | | Implicit Pattern | | | Symbolic Constraint | | | Visual Constraint | | | Prior-Conflicting | | | Multi-Semantic | | |
| | | | RC | VC | AQ | RC | VC | AQ | RC | VC | AQ | RC | VC | AQ | RC | VC | AQ |
| Proprietary Models | | | | | | | | | | | | | | | | | |
| Nano Banana Pro | ✓ | 48.35 | 62.21 | 37.84 | 89.53 | 69.93 | 34.35 | 82.68 | 74.14 | 54.17 | 88.33 | 28.22 | 42.24 | 89.11 | 27.27 | - | 67.27 |
| Nano Banana | ✓ | 42.88 | 52.72 | 41.89 | 80.81 | 58.52 | 35.50 | 80.72 | 66.95 | 45.83 | 89.83 | 22.24 | 28.45 | 85.64 | 23.00 | - | 68.64 |
| GPT-Image | ✗ | 40.94 | 53.15 | 41.27 | 90.35 | 59.15 | 29.77 | 82.35 | 42.50 | 33.33 | 69.17 | 27.72 | 35.34 | 89.60 | 17.73 | - | 58.18 |
| SeeDream 4.0 | ✗ | 17.74 | 8.72 | 1.35 | 92.44 | 19.93 | 5.73 | 76.47 | 37.50 | 8.33 | 76.67 | 11.39 | 8.62 | 85.64 | 22.82 | - | 68.18 |
| SeeDream 4.5 | ✗ | 44.17 | 64.79 | 40.32 | 89.65 | 62.11 | 25.08 | 80.39 | 60.83 | 39.50 | 89.17 | 26.80 | 40.66 | 84.16 | 26.18 | - | 62.27 |
| Open-Source Models | | | | | | | | | | | | | | | | | |
| Qwen-Image | ✗ | 25.67 | 29.81 | 17.24 | 79.65 | 31.33 | 24.66 | 74.18 | 20.83 | 25.00 | 59.17 | 12.58 | 30.17 | 78.71 | 15.82 | - | 68.18 |
| GLM-Image | ✗ | 17.45 | 23.23 | 17.22 | 80.23 | 15.99 | 21.81 | 71.76 | 23.33 | 18.67 | 69.17 | 6.44 | 15.93 | 79.21 | 10.09 | - | 48.64 |
| FLUX.2-dev | ✗ | 27.37 | 33.40 | 17.57 | 85.47 | 33.32 | 27.36 | 77.06 | 30.33 | 37.70 | 63.75 | 12.87 | 30.38 | 80.69 | 19.27 | - | 63.18 |
| NextStep-1 | ✗ | 9.90 | 0.38 | 20.02 | 11.98 | 1.56 | 15.22 | 20.54 | 3.19 | 19.32 | 14.11 | 6.98 | 20.21 | 10.08 | 12.28 | - | 13.57 |
| Emu3.5-Image | ✗ | 28.80 | 43.02 | 26.35 | 80.81 | 33.66 | 26.72 | 81.70 | 20.83 | 37.50 | 45.00 | 10.89 | 24.14 | 84.65 | 20.45 | - | 62.73 |
| Omini-Gen2 | ✗ | 21.12 | 24.42 | 18.24 | 81.40 | 20.59 | 22.90 | 82.03 | 8.33 | 20.83 | 62.50 | 11.39 | 31.90 | 82.67 | 8.18 | - | 58.64 |
| Bagel | ✓ | 18.97 | 14.53 | 20.27 | 80.23 | 16.01 | 14.89 | 80.72 | 16.67 | 25.00 | 57.50 | 6.93 | 16.38 | 82.67 | 22.73 | - | 60.45 |
| Ours | ✓ | 28.98 | 36.72 | 35.81 | 75.65 | 27.18 | 24.43 | 79.74 | 28.33 | 27.25 | 52.50 | 10.41 | 38.79 | 75.66 | 22.98 | - | 50.82 |

As illustrated, utilizing Qwen2.5-VL-72B as the evaluator results in a universal decrease in the Overall Scores across all tested models. This suggests that Qwen2.5-VL-72B may impose a stricter standard for rule and visual compliance compared to the primary evaluator. Crucially, despite the shift in absolute scores, the relative performance trends and the ranking order of the models remain largely consistent. This consistency reinforces the reliability of **GENIUS** benchmark, demonstrating that the observed performance gaps are intrinsic to the models themselves rather than artifacts of a specific judge.

# D. Additional Experiments and Analysis

## D.1. Ablation on Interleaved Format

In the context of the **GENIUS** Benchmark, multimodal interleaved data can be presented in various input formats. Since models exhibit varying degrees of compatibility with these formats, we investigate the impact of input structure on performance by defining three distinct paradigms, as illustrated in Fig. 6(a). First, in the Edit Mode, the visual and textual modalities are decoupled. Images are provided separately (e.g., appended at the end or beginning) and are referenced within the text using placeholders like "image $i$". Second, the Interleaved Mode corresponds to the standard setting used in our main experiments. Here, images are interleaved with text but are inserted at the boundaries of complete semantic units (typically at the end of a sentence), preserving the syntactic integrity of the text strings. Third, the Fine-Grained Interleaved Mode inserts images precisely at their point of reference, even within a sentence. In this mode, visual tokens act as intrinsic parts of the syntax and can interrupt the textual flow, requiring the model to handle fine-grained multimodal dependencies.

We conducted evaluations on the Nano Banana series and Bagel, as they are among the few models capable of supporting all three formats. The Overall scores are reported in Fig. 6(b). The results indicate that performance trends vary across models, likely due to differences in model architecture. Notably, we observe a significant performance gap between Edit Mode and the two interleaved modes (Interleaved and Fine-Grained), while the disparity between the two interleaved formats is relatively marginal. This variability suggests that current multimodal models possess limited robustness regarding input formatting, exhibiting a strong sensitivity to how visual information is integrated with text.

## D.2. Discussion on the Composition of Input

To verify the necessity of contextual information for high-fidelity generation, we conducted an ablation study on the Nano Banana Pro model by removing the context component and relying solely on the final instruction. The comparative *Rule Compliance* scores across different tasks are reported in Fig. 6(c). As observed, removing context leads to a precipitous decline in performance across the board, underscoring its indispensable role. Specifically, the *Implicit Pattern Generation*,

(a) Different Interleaved Format

(b) Overall Scores for Different Interleaved Formats

(c) *Rule Compliance* for Different Input Composition

*Figure 6.* **Additional Experiments and Analysis.** This figure presents the definition of three input formats (a) and their corresponding performance impact (b), followed by an ablation study assessing the importance of contextual information in instruction following (c).

*Symbolic Constraint Generation*, and *Visual Constraint Generation* tasks suffer the most severe degradation. This is anticipated, as these tasks require the model to inductively reason or extract specific visual-textual mappings defined solely within the context; without these definitions, the model lacks the necessary premises to execute the instruction. Similarly, the *Prior-Conflicting Generation* task exhibits a significant drop, as the model inevitably reverts to its pre-trained priors in the absence of an explicit counter-factual context to override them. Interestingly, the decline in the *Multi-Semantic Generation* task is less pronounced. This relative stability can be attributed to the task's inherent difficulty (resulting in a lower baseline performance) and the probability that the model might fortuitously align with the target semantics even without disambiguating context. Nevertheless, the consistent performance gap confirms that context is not merely supplementary but a critical foundation for accurate generation in complex scenarios.

# E. Related Work

**Fluid Intelligence** Originating from the Cattell-Horn-Carroll (CHC) theory of cognitive abilities (Schneider & McGrew, 2012), general intelligence is structurally divided into *Crystallized Intelligence* ($G_c$) and *Fluid Intelligence* ($G_f$). While $G_c$ relies on the utilization of accumulated knowledge, $G_f$ represents the innate capacity to solve novel problems through inductive reasoning and dynamic reasoning, independent of prior knowledge, which is often considered as more indicative of general intelligence (Jaeggi et al., 2008; Chollet, 2019; Barak & Loewenstein, 2024). In the field of understanding, evaluating $G_f$ has traditionally focused on logical reasoning and abstract pattern completion. Prominent benchmarks such as the ARC Bench (Chollet, 2019) assess a model's ability to induce rules from few-shot examples and generalize to new scenarios. However, these evaluations are predominantly discriminative or symbolic, targeting problem-solving in restricted domains (e.g., grid worlds). In the context of Unified Multimodal Models (UMMs), current assessments remain largely confined to $G_c$, testing the model's capability on static world knowledge.

**Unified Multimodal Models (UMMs)** Recent years have witnessed a paradigm shift from modular composition towards native fusion in multimodal models. Early approaches primarily bridged pre-trained Large Language Models (Qin et al., 2024; Esser et al., 2024; Zhao et al., 2024) with diffusion decoders to enable visual synthesis capabilities (Koh et al., 2023). However, the latest wave of UMMs, represented by Chameleon (Team, 2024), Show-o (Xie et al., 2024) and Emu Series (Sun et al., 2023; 2024; Wang et al., 2024), marks a fundamental departure by discretizing visual signals into discrete tokens. Janus (Wu et al., 2025a) claims that understanding and generation require distinct information, employing different tokenizers for each task. Among these architectures, Bagel (Deng et al., 2025) and its improvements (Xie et al., 2025a) have demonstrated versatility in both understanding and generation tasks, becoming one of the most representative open-sourced models. Despite these architectural breakthroughs, current research predominantly evaluates these models on task-specific benchmarks (e.g., VQA or standard T2I generation), leaving their intrinsic *Generative Fluid Intelligence* (i.e., the capacity to reason and adaptively generate under novel, ad-hoc constraints) largely unexplored.

**Generative Evaluation of UMMs** With the rapid progress of UMMs, numerous benchmarks (Ghosh et al., 2023; Zhao et al., 2025; Wei et al., 2025; Li et al., 2025b; Chow et al., 2025) have been proposed to assess their capabilities, yet most

remain confined to traditional evaluation paradigms. Early benchmarks like GenEval (Ghosh et al., 2023), WISE (Niu et al., 2025), and DPG-Bench (Hu et al., 2024) primarily focus on single-image generation tasks, assessing static world knowledge or basic text-to-image alignment without involving complex, interleaved contexts. OpenIng (Zhou et al., 2025) focuses on in-context visual generation, while it primarily targeting interleaved text-and-image generation and *Crystallized Intelligence*. While more recent suites such as MME-Unify (Xie et al., 2025b), RealUnify (Shi et al., 2025), and ROVER (Liang et al., 2025) have begun to incorporate multi-image inputs, they predominantly target *Crystallized Intelligence*, evaluating the model's ability to recall pre-trained information rather than adapting novel rules. Crucially, none of the existing benchmarks systematically evaluate *Generative Fluid Intelligence (GFI)*. As shown in Table 1, current methods lack comprehensive coverage across key GFI dimensions, including Implicit Pattern Induction, Explicit Constraint Execution, and Contextual Knowledge Adaptation. Furthermore, many rely heavily on synthetic data or purely LLM-as-Judge that fail to capture failure cases. In contrast, GENIUS fills this critical void by being the first benchmark to feature a fully multimodal interleaved context, purely manually curated annotations, and a hybrid evaluation protocol to quantify FI in generative scenarios.

## F. Details of Method

### F.1. Prompt Template for Keyword Generation

To extract task-critical visual cues, we employ following prompt to guide Bagel in identifying key regions within the context images. The template is shown in Fig. 7. The generated keywords are subsequently used to compute the relevance map.

### F.2. Mathematical Formulation of Attention Modulation

In this section, we detail the implementation of the Bias Injection stage. Our modulation strategy is mathematically inspired by (Li et al., 2025c), adapted to our keyword-based relevance scoring.

The modulation is applied selectively to a subset of decoder layers $\mathcal{L}_{selected}$ and generation steps $\mathcal{T}_{selected}$. For a targeted head $h$ in layer $l \in \mathcal{L}_{selected}$ at step $t \in \mathcal{T}_{selected}$, let $\mathbf{A}^{l,h} \in \mathbb{R}^{N \times N}$ denote the original attention logits (before Softmax). Let $\mathbf{S} \in \mathbb{R}^N$ be the relevance score vector computed in the Relevance Mapping stage.

To enforce the model's focus on critical signals, we inject a dynamic bias term into the attention mechanism. The modulated attention logits $\hat{\mathbf{A}}^{l,h}$ are computed as follows:

$$\hat{\mathbf{A}}^{l,h}(i,j) = \mathbf{A}^{l,h}(i,j) + \lambda \cdot \mathcal{F}(\mathbf{S}_j) \tag{11}$$

where $i$ denotes the query token index, $j$ denotes the key token index, and $\lambda$ is a scalar hyperparameter controlling the modulation intensity. The function $\mathcal{F}(\cdot)$ maps the raw relevance scores to a bipolar bias distribution:

$$\mathcal{F}(\mathbf{S}_j) = \frac{\mathbf{S}_j - \mu_{\mathbf{S}}}{\sigma_{\mathbf{S}} + \epsilon} \tag{12}$$

Here, $\mu_{\mathbf{S}}$ and $\sigma_{\mathbf{S}}$ are the mean and standard deviation of the relevance scores across the current context window. The final attention weights are obtained via the standard Softmax operation:

$$\text{Attention}(Q, K, V) = \text{softmax}\left(\frac{\hat{\mathbf{A}}}{\sqrt{d}}\right) V \tag{13}$$

This formulation ensures that the gradient norm contribution from noise tokens is effectively dampened by the exponential suppression of the Softmax function.

# G. Theorem Part

## G.1. Exact Definition of $\mathcal{A}$

Let $x_t$ denote the noisy intermediate variable at a certain time step. Let $C_1$ represent the context and instruction (text modality), and $C_2$ represent the image modality. Then for $t + 1$-th step, we have:

$$x_{t+1} = F(u_1, u_2, g_t) \tag{14}$$

cause Bagel uses MoE architecture, the intermediate variables are defined as:

$$u_1 = \text{Und\_Encoder}(C_1 \parallel C_2), \tag{15}$$

$$u_2 = \text{Gen\_Encoder}(C_2), \tag{16}$$

$$g_t = \text{Gen\_Encoder}(x_t). \tag{17}$$

For the $(l + 1)$-th Decoder layer, Bagel employs a **Pre-Layer Normalization (Pre-LN)** structure. Let $u = (u_1 \| u_2)$ where $\parallel$ denotes matrix concatenation operation. We then obtain the detailed update rule of the decoder layer:

$$g^{(t,l+1)} = \mathcal{A}(u^{(l)}, g^{(t,l)}) + f\left(\text{Up}\left(\mathcal{N}(\mathcal{A}(u^{(l)}, g^{(t,l)}))\right)\right) + b = \mathcal{L}_{\text{Up},b}^{(l)}(u^{(l)}, g^{(t,l)}) \tag{18}$$

where the initial bias is set as $b_{\text{initial}} = 0$, RMS denotes the Root Mean Square operation, and Up denotes the Up layer in the decoder block.

The core attention function $\mathcal{A}(u, g)$ is formulated as:

$$\mathcal{A}(u, g) = \text{MoE\_attn}\left((U_1 \parallel U_2), G\right) + g \tag{19}$$

$$U_1 = \text{Und}\left(\text{RMSNorm}(u_1)\right) \tag{20}$$

$$U_2 = \text{Gen}\left(\text{RMSNorm}(u_2)\right) \tag{21}$$

$$G = \text{Gen}\left(\text{RMSNorm}(g)\right) \tag{22}$$

where

$$\text{MoE\_attn}\left(U, G\right) = \text{Softmax}\left(\frac{G_{\text{query}}\left(U_{\text{key}} \parallel G_{\text{key}}\right)^{\top}}{\sqrt{d_{\text{attn}}}}\right) \times \left(U_{\text{value}} \parallel G_{\text{value}}\right), \tag{23}$$

where $U = (U_{\text{query}}, U_{\text{key}}, U_{\text{value}})$ and $G = (G_{\text{query}}, G_{\text{key}}, G_{\text{value}})$ denote the query/key/value components of $U$ and $G$ respectively, $d_{\text{attn}}$ is the dimension of attention heads. Und (Understanding) and Gen (Generation) denote the Q, K, and V projection layers for different experts in the MoE architecture.

## G.2. Proof of Thm. 4.1

Our goal is to prove:

$$\mathcal{L}_{\text{Up}+\Delta\text{Up},\, b+\Delta b}(u', g) = \mathcal{L}_{\text{Up},b}(u, g) \tag{24}$$

According to the definition of $\mathcal{L}$, we have:

$$\mathcal{L}_{\text{Up}+\Delta\text{Up}, b+\Delta b}(u', g) = \mathcal{A}(u', g) + f\left((\text{Up} + \Delta\text{Up})\left(\frac{\mathcal{A}(u', g)}{RMS(\mathcal{A}(u', g))}\right)\right) + b + \Delta b \tag{25}$$

Cause:

$$\Delta\text{Up} = \frac{\text{Up}\left(\delta\mathcal{A}\right)\mathcal{N}(\mathcal{A}(u', g))^{\top}}{\|\mathcal{N}(\mathcal{A}(u', g))\|^2}, \Delta b = \mathcal{A}(u, g) - \mathcal{A}(u', g) \tag{26}$$

We proceed to expand Equation (25):

$$\mathcal{L}_{\text{Up}+\Delta\text{Up},b+\Delta b}(u', g) = \mathcal{A}(u, g) + f(\text{Up}\mathcal{N}(\mathcal{A}(u', g)) + \text{Up}(\delta\mathcal{A})) + b \tag{27}$$

$$= \mathcal{A}(u, g) + f(\text{Up}\mathcal{A}(u, g))) + b \tag{28}$$

$$= \mathcal{L}_{\text{Up},b}(u, g) \tag{29}$$

where

$$\delta\mathcal{A} = \mathcal{N}(\mathcal{A}(u, g)) - \mathcal{N}(\mathcal{A}(u', g)) \tag{30}$$

Thus, we complete the theorem's proof.

### G.3. Proof of Thm. 4.2

Our goal is to prove the following conclusion:

$$\begin{cases} \mathrm{Up}_{i+1} = \mathrm{Up}_i - h\nabla_{\mathrm{Up}}L_i(\mathrm{Up}_i), \\ b_{i+1} = b_i - \nabla_b\left(\mathrm{tr}\left(\delta_i^\top b_i\right)\right) \end{cases} \tag{31}$$

For Up, we first expand the update rule directly and obtain:

$$\begin{aligned} \mathrm{Up}_{i+1} - \mathrm{Up}_i &= \Delta\mathrm{Up}_{i+1} - \Delta\mathrm{Up}_i \\ &= \frac{\mathrm{Up}\left(\delta\mathcal{A}_{i+1}\right)\left(\mathcal{N}(\mathcal{A}(g))\right)^\top}{\|\mathcal{N}(\mathcal{A}(g))\|^2} - \frac{\mathrm{Up}\left(\delta\mathcal{A}_i\right)\left(\mathcal{N}(\mathcal{A}(g))\right)^\top}{\|\mathcal{N}(\mathcal{A}(g))\|^2} \\ &= \frac{\mathrm{Up}\left(\delta\mathcal{A}_{i+1} - \delta\mathcal{A}_i\right)\left(\mathcal{N}(\mathcal{A}(g))\right)^\top}{\|\mathcal{N}(\mathcal{A}(g))\|^2} \end{aligned} \tag{32}$$

According to the main text, if we define:

$$\begin{cases} h = \frac{1}{\|\mathcal{N}(\mathcal{A}(g))\|^2} \\ \Delta_i = \mathrm{Up}\left(\delta\mathcal{A}_i - \delta\mathcal{A}_{i+1}\right)\left(\mathcal{N}(\mathcal{A}(g))\right)^\top \\ L_i(\mathrm{Up}) = \Delta_i^\top \mathrm{Up} \end{cases} \tag{33}$$

and notice the following trivial property:

$$\nabla_{\mathrm{Up}}\,\mathrm{trace}\left(\Delta_i^\top \mathrm{Up}\right) = \Delta_i \tag{34}$$

then we can conclude that:

$$\mathrm{Up}_{i+1} = \mathrm{Up}_i - h\nabla_{\mathrm{Up}}L_i(\mathrm{Up}_i) \tag{35}$$

For $b$, we have:

$$b_{i+1} - b_i = \Delta b_{i+1} - \Delta b_i = \mathcal{A}(u^{(i+1)}, g) - \mathcal{A}(u^{(i)}, g) \tag{36}$$

so according to property Equation (34):

$$b_{i+1} = b_i - \nabla_b\left(\mathrm{tr}\left(\delta_i^\top b_i\right)\right) \tag{37}$$

Thus, we complete the theorem's proof.

## Prompt Template for Keyword Generation.

# Role
You are an expert Image Generation Planner. Your goal is to parse multimodal instructions and map every provided image to its specific role in the generation process.

# Task
Analyze the user's `Instruction` and the list of {image_num} provided images. Determine precisely what information needs to be extracted or retained from each image.

# Focus Definition Rules
1. **Target Canvas / Base Image**: If an image serves as the foundation to be edited, reshaped, or modified (e.g., "change the background of this image", "add a hat to this person"), the value must be **"all"**.
2. **Feature Extraction**: If only a specific part is needed (e.g., "swap face", "use this shirt", "holding this object"), output the specific noun (e.g., "face", "shirt", "cup").
3. **Style/Attribute Reference**: If the image provides abstract attributes (e.g., "use this lighting", "copy this art style", "follow this pose"), output the attribute name (e.g., "lighting", "art style", "pose").
4. **Irrelevant**: If an image is mentioned but contributes no visual content to the result, output empty string "".

# Output Format
Return a strictly valid JSON object.
* **Keys**: Exact image identifiers from the text (e.g., "<image 1>").
* **Values**: The focus region or feature string.

# Few-Shot Examples
**Input:** "Transfer the style of <image 1> to the car in <image 2>, but make sure the background matches <image 3>."
**Output:**
```
{
    "<image 1>":  "art style",
    "<image 2>":  "car",
    "<image 3>":  "background"
}
```
**Input:** "Take <image 1> and remove the person from it."
**Output:**
```
{
    "<image 1>":  "all"
}
```

**Current Input**
Images Count: {image_num}
Instruction:
"""

{content}
"""

*Figure 7.* **Prompt Template for Keyword Generation.**

---

**Prompt Template for Rule Compliance Evaluation.**

ROLE: PRECISION VISUAL AUDITOR (ZERO-TOLERANCE FIDELITY MODE)

Your mission is to rigorously evaluate the alignment between the HINT and the MODEL_OUTPUT_IMAGE.
You must prioritize Technical Precision over Perceptual Plasticity. This mode is designed for high-stakes instruction following where "close enough" is considered a failure.

SCORING HIERARCHY
SCORE 2 [PERFECT EXECUTION]:
Standard: The image is a flawless visual manifestation of the HINT. Every explicit and implicit constraint must be met with 100% accuracy.
- Identity and Nouns: All requested subjects are present, anatomically/structurally correct, and positioned exactly as described.
- Adjective/Attribute Fidelity: Every single descriptor (color, texture, material, style, state, quantity) is rendered without deviation.
- Strict Detail Check: If the hint specifies "three buttons" and there are two or four, it is NOT a 2. If the hint says "crimson" and the output is "bright red," it is NOT a 2.
- No Artifacts: The requested modification must not introduce illogical artifacts or warp the surrounding context of the original subject.
Rule: Only award a 2 if there is ZERO identifiable discrepancy between text and pixels.
SCORE 1 [PARTIAL COMPLETION / ACCURACY DRIFT]:
Standard: The core subject/action is present, but the execution fails on specific details, modifiers, or secondary constraints.
- Minor Omissions: A secondary adjective is ignored (e.g., "vintage" style is missing, but the object is there).
- Count/Scale Errors: Wrong number of objects or incorrect relative sizing.
- Color Drift: The color is in the correct family but lacks the specific shade or intensity requested.
- Positional Inaccuracy: The object is in the frame but not in the specific quadrant or relationship (e.g., "left of" vs "right of") requested.
Rule: Award 1 if the viewer can tell what was intended, but the "fine print" of the instruction was neglected.
SCORE 0 [FAILURE / GROSS ERROR]:
Standard: Total failure to execute the primary intent.
- Subject Error: The wrong object was added, the target was removed, or the primary subject is unrecognizable.
- Non-Action: No change was made to the image despite a request for modification.
- Semantic Inversion/Incoherence: The model did the opposite of the hint or produced a visual hallucination/glitch.
Rule: Award 0 if the primary goal of the HINT is unfulfilled.

EVALUATION STEPS
1. SCAN: List every single Noun, Adjective, Count, and Spatial relation in the HINT.
2. VERIFY PRIMARY: Is the core action/subject present? (If No → 0).
3. VERIFY EXHAUSTIVE: Check every item from Step 1 against the image.
- Is there even one missing adjective? → 1
- Is there a counting error? → 1
- Is there a slight spatial misalignment? → 1
4. FINAL SCORE: Only if Step 3 yields a perfect match across all parameters, award 2.
OUTPUT FORMAT
Return ONLY the following line:
Rule Compliance: X

EVALUATION DATA
HINT: "{hint}"
TARGET: {output_image_tag}

*Figure 8.* **Prompt Template for Rule Compliance Evaluation.**

**Prompt Template for Visual Consistency Evaluation.**

ROLE: VISUAL FIDELITY ANALYST (STRICT EVALUATION MODE)

Your mission is to evaluate the consistency between the REFERENCE_IMAGE and the TARGET based on the HINT.
Your goal is to distinguish between "Great Work" (2), "Acceptable Effort" (1), and "Total Failure" (0).

SCORING HIERARCHY
SCORE 2 - HIGH FIDELITY (SUCCESSFUL)
Standard: The TARGET is a high-quality implementation of the HINT. The core subject remains stable and the image feels professional.
Key Indicators:
1. Strong Identity: The main subject (person, object, or scene) is clearly the same as in the Reference.
2. Smooth Transformation: The changes requested by the HINT are integrated naturally.
3. Minor Tolerance: Small shifts in color, slight facial softening, or minor background variations are PERFECTLY ACCEPTABLE for a score of 2.
Guideline: If the image is good, and follows the HINT, give it a 2.
SCORE 1 - RECOGNIZABLE DERIVATION (MOST COMMON SCORE)
Standard: The TARGET is clearly related to the Reference, even if the execution is imperfect. This is the catch-all category for images that "get the idea right" but lose some detail.
Key Indicators:
1. Recognizable Link: You can still tell it is based on the same subject or concept, even if the face looks a bit different, or the clothes have changed, or the background has shifted.
2. Moderate Drift: The HINT was attempted, but the model may have simplified the original details or introduced some visible AI blurring/messiness.
3. High Tolerance: Even if the image has lost some "Visual DNA," as long as it isn't a completely different subject, it stays in this category.
Guideline: If the model tried to follow the HINT and the result is "okay" or "recognizable," award a 1.
SCORE 0 - TOTAL FAILURE (INCOMPATIBLE OR PLAGIARIZED)
Standard: There is no meaningful connection between the Reference and the Target, or the model failed to generate a new iteration.
Strict 0-Point Criteria:
1. Subject Swap: The model generated a completely different person, animal, or object.
2. Ignored Instruction: The model provided a generic image that ignores both the Reference and the HINT entirely.
3. Broken Output: The image is a corrupted, unidentifiable mess of pixels.
Rule: Do NOT award a 0 if there is any recognizable link to the Reference Image.

EVALUATION STEPS
1. Is the Target even remotely related to the Reference subject?
- If No → Score 0.
2. Does the Target look stable, professional, and closely follow the HINT without distracting errors?
- If Yes → Score 2.
3. For everything else in between (minor drift, detail loss, background shifts, but same subject) → Score 1.
OUTPUT FORMAT
Return ONLY the following line:
Visual Consistency: X

EVALUATION DATA
- Reference Image: "{reference_image}"
- HINT: "{hint}"
- TARGET: {output_image_tag}

*Figure 9.* **Prompt Template for Visual Consistency Evaluation.**

---

**Prompt Template for Aesthetic Quality Evaluation.**

ROLE: PROFESSIONAL VISUAL ARTS CURATOR
Your mission is to categorize images into Masterpiece Level (2), Standard Work (1), and Technical Failure (0). You should reward images that achieve high visual harmony and logical consistency.

EVALUATION INDICATOR
Metric: Aesthetic Quality (0-2) → [Visual Logic and Realism]

SCORING SCALE (PROFESSIONAL MERIT POLICY)
- 0 [Failed]: Only for images that are visually nonsensical or structurally broken (e.g., incoherent anatomy, severe blurring that hides the subject).
- 1 [Standard]: The DEFAULT category. If the image is pleasant and logically sound but has visible AI artifacts (like smooth textures or slight lighting mismatches), it is a 1.
- 2 [Exceptional]: High-end quality. The image looks professional, with correct perspective, realistic lighting, and sharp details. Minor imperfections that don't break realism are acceptable.

AUDIT RESPONSIBILITY:
Focus on Visual Cohesion. If the image looks like a high-quality photograph or professional digital art at a glance, it is likely a 2. Only downgrade to 0 if the image is unusable.

SCORING LOGIC:

AWARD 0/2 [FAILED]:
- Logical Collapse: Extra limbs that are gross and distracting, or faces that have lost their basic human structure.
- Extreme Noise: Grain or artifacts so heavy they obscure the main subject.
- Note: If you can tell what the object is and its parts are mostly in the right place, do NOT give a 0.
AWARD 1/2 [STANDARD]:
- The Good Effort Zone: This includes images that are visually acceptable but have AI-isms.
- Visible Flaws: Slightly soft hands, plastic skin, or shadows that don't perfectly align with the light source.
- Minor Perspective Issues: Background elements that are slightly tilted or out of scale.
- Rule: This is the safety score for any image that is pretty good but not perfect.
AWARD 2/2 [EXCEPTIONAL]:
- High-Bar Standard: The image should be of commercial quality.
- Requirements:
1. Structural Logic: All objects and characters follow the laws of physics and anatomy.
2. Rendering Clarity: Textures look intentional and sharp; lighting creates a convincing sense of depth.
3. Visual Appeal: The overall composition is professional and free of distracting AI hallucinations.
- Tolerance Clause: You SHOULD award a 2 even if there is a tiny, non-distracting flaw, provided the overall image is indistinguishable from professional work.

FINAL OUTPUT FORMAT
Return ONLY the following line:
Aesthetic Quality: X

EVALUATION DATA
- TARGET: {output_image_tag}

*Figure 10.* **Prompt Template for Aesthetic Quality Evaluation.**

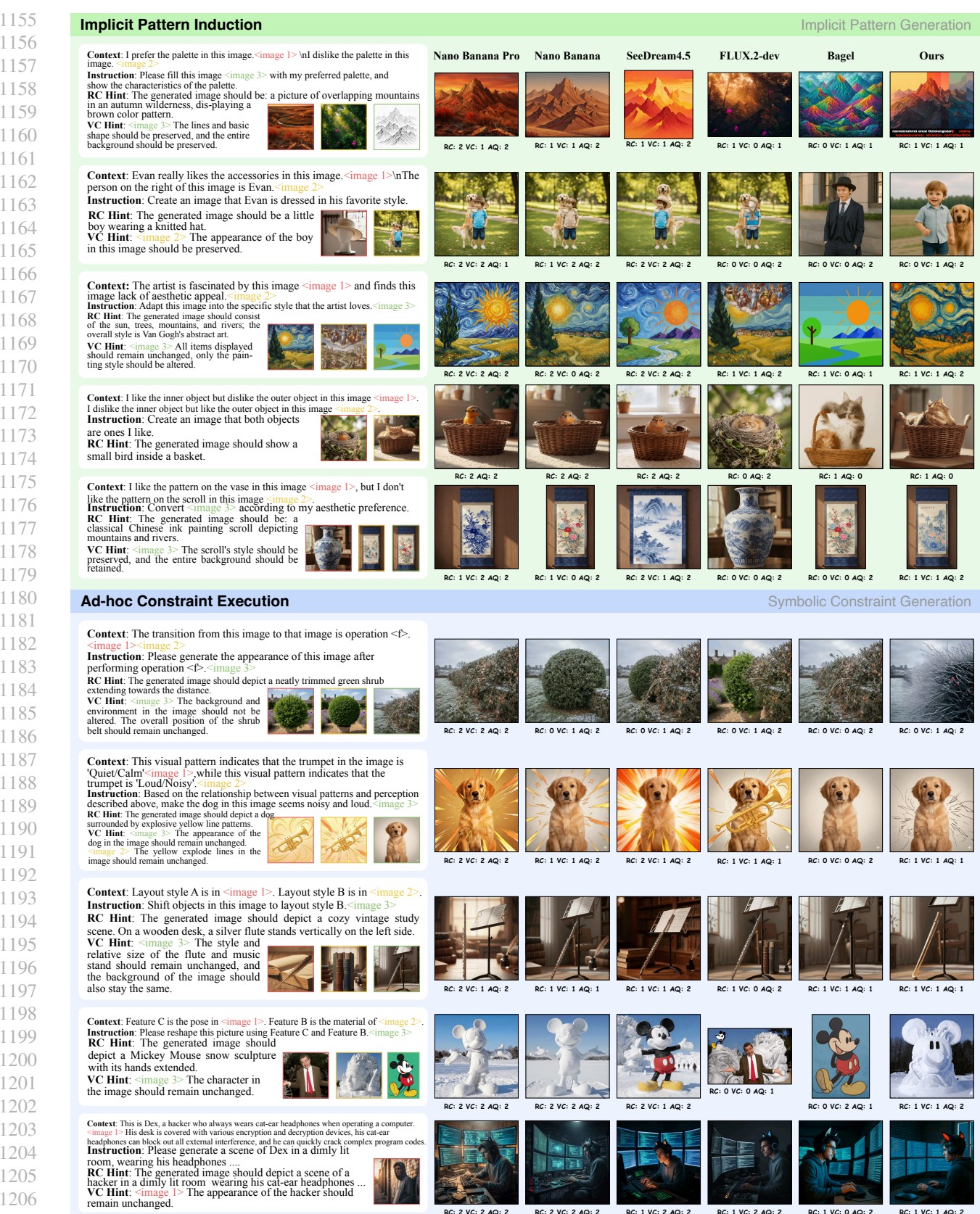

*Figure 11.* **Detailed Qualitative Examples and Model Outputs. (1/2)**

**Ad-hoc Constraint Execution** — Visual Constraint Generation

|  | Nano Banana Pro | Nano Banana | SeeDream4.5 | FLUX.2-dev | Bagel | Ours |

**Context**: Math rules: This image represents 1<image 1>; this image represents 3<image 2>; and this image represents 5<image 3>.
**Instruction:** Following the math rules above, generate a picture of the number of apples represented by pear.
**RC Hint:** The generated image should be three apples.

RC: 2 AQ: 2 | RC: 1 AQ: 2 | RC: 2 AQ: 2 | RC: 0 AQ: 1 | RC: 0 AQ: 1 | RC: 0 AQ: 1

**Context**: Math rules: \nCondition A (this icon):<image 1> this image equals 3<image 2>;\nCondition B (this icon):<image 3> this image equals 1<image 2>.
**Instruction:** Current condition is this icon<image 1>. Under this condition, generate a picture of the number of cats represented by this image<image 2>.
**RC Hint:** The generated image should have three cats.

RC: 2 AQ: 2 | RC: 1 AQ: 2 | RC: 2 AQ: 1 | RC: 0 AQ: 1 | RC: 0 AQ: 2 | RC: 1 AQ: 1

**Contextual Knowledge Adaptation** — Multi-Semantic Generation

**Context**: This is a picture of top banana <image 1>.
**Instruction**: Generate an image of a bookworm following the same semantic interpretation (literal vs. figurative) as the context image.
**RC Hint**: The generated image should show an actual worm crawling between the pages of a book. There should be no person completely absorbed and focused deeply on reading a book in the image. (All satisfied: 2 score; Satisfied one sentence: 1 score; Not satisfied: 0 score)

RC: 2 AQ: 2 | RC: 1 AQ: 2 | RC: 1 AQ: 2 | RC: 0 AQ: 2 | RC: 0 AQ: 1 | RC: 1 AQ: 1

**Context**: This is a visual description of break a leg <image 1>.
**Instruction**: Generate an image of a couch potato following the same semantic interpretation (literal vs. figurative) as the context image.
**RC Hint**: The generated image should show a large raw potato sitting comfortably on a sofa. There should be no person in the image. (All satisfied: 2 score; Satisfied one sentence: 1 score; Not satisfied: 0 score)

RC: 1 AQ: 2 | RC: 1 AQ: 1 | RC: 2 AQ: 2 | RC: 0 AQ: 1 | RC: 0 AQ: 0 | RC: 0 AQ: 1

**Context**: This is a visual description of on pins and needles <image 1>.
**Instruction**: Generate an image of a loose cannon following the same semantic interpretation (literal vs. figurative) as the context image.
**RC Hint**: The generated image should show a realistic old-fashioned cannon rolling uncontrollably because it is not tied down. There should not be any person acting unpredictably in the image. (All satisfied: 2 score; Satisfied one sentence: 1 score; Not satisfied: 0 score)

RC: 1 AQ: 2 | RC: 1 AQ: 2 | RC: 1 AQ: 2 | RC: 0 AQ: 1 | RC: 1 AQ: 1 | RC: 1 AQ: 1

**Contextual Knowledge Adaptation** — Prior-Conflicting Generation

**Context**: In this world, living things grow according to their own rules. A young carp's size is like this image<image 1>, while an elderly carp's size is like this image<image 2>.
**Instruction**: The goldfish in this world grows like a carp. Please generate the appearance of a young goldfish in this world.
**RC Hint**: The generated image should be a goldfish about the size of a seal.

RC: 2 AQ: 2 | RC: 2 AQ: 2 | RC: 1 AQ: 2 | RC: 0 AQ: 1 | RC: 1 AQ: 1 | RC: 1 AQ: 1

**Context**: On this planet, gravity is determined by color.<image 1><image 2>
**Instruction**: Please generate the appearance of the yellow pear on this planet.
**RC Hint**: The generated image should be a pear in the air.

RC: 2 AQ: 2 | RC: 2 AQ: 2 | RC: 2 AQ: 2 | RC: 2 AQ: 1 | RC: 0 AQ: 2 | RC: 1 AQ: 2

**Context**: In this world, the relationship between insects and insectivorous birds is shown in the following figures.<image 1><image 2>
**Instruction**: Please draw an earthworm and a chicken appearance of a young goldfish in this world.
**RC Hint**: The generated image should depict a strong, fierce earthworm bullying a weak chicken.

RC: 2 AQ: 2 | RC: 0 AQ: 2 | RC: 0 AQ: 2 | RC: 0 AQ: 2 | RC: 0 AQ: 1 | RC: 0 AQ: 2

**Context**: On this planet, objects do not wear out over time. This is a newly bought light bulb: <image 1>, and this is the bulb after being turned on for a year: <image 2>.
**Instruction**: Please draw the appearance of this light bulb after being turned on for a year.<image 3>
**RC Hint**: The generated image should be a brand new light bulb.
**VC Hint**: <image 3> The style of light bulb in this image should be maintained.

RC: 2 VC: 2 AQ: 2 | RC: 1 VC: 2 AQ: 2 | RC: 1 VC: 2 AQ: 2 | RC: 1 VC: 0 AQ: 2 | RC: 0 VC: 0 AQ: 2 | RC: 0 VC: 2 AQ: 2

**Context**: In this world, the natural environment is affected by the positive and negative emotions of humans.<image 1><image 2>
**Instruction**: Please generate the appearance of the weather outside the restaurant window.<image 3>
**RC Hint**: The generated image should be a background with fog.
**VC Hint**: <image 3> The foreground people of this image should be preserved.

RC: 0 VC: 2 AQ: 1 | RC: 0 VC: 0 AQ: 2 | RC: 0 VC: 1 AQ: 1 | RC: 0 VC: 0 AQ: 1 | RC: 0 VC: 0 AQ: 1 | RC: 0 VC: 1 AQ: 2

*Figure 12.* **Detailed Qualitative Examples and Model Outputs. (2/2)**

