# OpenReview forum: "GENIUS: Generative Fluid Intelligence Evaluation Suite"
_ICML.cc/2026/Conference — Submitted to ICML 2026_

### Official Review · Reviewer_GZsL · 2026-03-12

**Soundness:** 3
**Presentation:** 4
**Significance:** 3
**Originality:** 3
**Overall Recommendation:** 4
**Confidence:** 2

**Summary:**

This paper introduces GENIUS, a benchmark evaluating Generative Fluid Intelligence (GFI) in multimodal models. It rigorously tests models on implicit pattern induction, ad-hoc constraint execution, and contextual knowledge adaptation. Finding significant deficits in 12 representative models , it proposes a training-free attention adjustment mechanism to improve GFI capabilities.

**Compliance With Llm Reviewing Policy:**

Affirmed.

**Key Questions For Authors:**

Refer to the above.

**Limitations:**

Refer to the above.

**Strengths And Weaknesses:**

**Soundness**: The evaluation protocol is technically robust, employing a hybrid LMM-as-a-judge system validated against human experts. The validation shows an exceptionally strong global consistency, achieving a Pearson correlation of r=0.9630 for Nano Banana Pro and r=0.9659 for Bagel. Furthermore, the mathematical derivation linking in-context learning to implicit gradient descent is rigorous. However, the proposed attention modulation relies heavily on the model's intrinsic keyword distillation capability. Failing this initial semantic reasoning step could easily propagate errors to the downstream bias injection and relevance mapping.

**Presentation**: The paper is exceptionally well-structured, successfully translating the Cattell-Horn-Carroll cognitive theory into three concrete GFI primitives. Detailed prompt templates and visual examples in the appendices greatly aid reproducibility. Nevertheless, the narrative transition in Section 4.2 from the continuous mathematical derivations of implicit gradient descent to the discrete, keyword-based attention bias feels slightly abrupt and could be bridged more smoothly.

**Significance**: This work addresses a critical benchmark bottleneck by shifting focus from Crystallized Intelligence to dynamic reasoning and adaptation. Exposing the illusion of competence—where models achieve high aesthetic quality but fail logical constraints—is highly valuable. However, the proposed training-free baseline's impact is somewhat modest, boosting Bagel's overall score. This suggests that intrinsic architectural deficits in GFI might necessitate fundamental training paradigm shifts rather than mere inference-time interventions.

**Originality**: Formalizing GFI for visual generation fills a significant theoretical void. Deliberately excluding static prior knowledge to strictly evaluate ad-hoc reasoning is a highly original departure from standard benchmarks. While the attention re-weighting mechanism explicitly suppressing noise tokens shares similarities with existing attention-steering techniques, framing it through the lens of implicit gradient descent provides a novel and rigorously justified application.

---

> ### Author Rebuttal · Authors · 2026-03-30
>
> We thank the **reviewer GZsL** for the thoughtful feedback and constructive suggestions. We address the key concerns below.
>
> > **W1**: The concern of heavy reliance on the model's intrinsic keyword distillation capability
>
> We contend that keyword extraction is unlikely to become a system bottleneck. Current Unified Multimodal Models (UMMs, e.g. Bagel) have achieved semantic understanding performance comparable to specialized understanding-only models (Qwen2.5-VL 7B)[1]. As pre-training techniques continue to evolve, the accuracy of such reasoning tasks will further strengthen. Consequently, keyword distillation serves as a highly reliable foundation for subsequent attention adjustment.
>
> > **W2**: Derivations slightly abrupt.
>
> We appreciate the feedback. In the final version, we will strengthen the logical bridge between the mathematical derivations in Section 4.2 and the keyword-based implementation in Section 4.3. A transitional paragraph will be added to map the theoretical gradients to the semantic guidance, ensuring a smoother and more coherent narrative flow.
>
> > **W3**: Training-Free Baseline's impact.
>
> We clarify that our baseline is intended as a "possible view" for diagnosing GFI deficits. The primary value of GENIUS lies in "exposing limitations", demonstrating that inference-time optimizations alone cannot fully bridge deep architectural gaps. Our baseline serves as an "existence proof" that latent GFI potential can be activated, ultimately highlighting the necessity of a fundamental training paradigm shift.
>
> [1] Emerging Properties in Unified Multimodal Pretraining

---

> > ### Author Rebuttal · Reviewer_GZsL · 2026-04-05
> >
> > The authors' response has effectively mitigated my concerns. Consequently, I will maintain my original positive recommendation.

---

> > > ### Author Response · Authors · 2026-04-06
> > >
> > > Thank you very much for reading our rebuttal so carefully. We are glad that the additional clarifications and experiments helped resolve your concerns. Your feedback has been very valuable in improving both the clarity and the rigor of the paper, and we will make sure these points are fully reflected in the final version.

---

### Official Review · Reviewer_2Sop · 2026-03-13

**Soundness:** 3
**Presentation:** 3
**Significance:** 3
**Originality:** 3
**Overall Recommendation:** 4
**Confidence:** 3

**Summary:**

The paper proposes Generative Fluid Intelligence (GFI) as a way to characterize context-dependent visual generation, and defines it through three primitives: implicit pattern induction, abstract dynamic reasoning under ad-hoc constraints, and adaptive inhibition / contextual knowledge adaptation. To study this, the authors introduce GENIUS, a benchmark of 510 expert-curated examples across 20 sub-tasks, together with a hybrid evaluation protocol that measures rule compliance, visual consistency, and aesthetic quality.

The paper evaluates 12 multimodal generation models on this benchmark, including both open-source and proprietary systems, and also includes diagnostic experiments that compare direct generation with prompting-based and VQA-style variants to better understand where failures arise.

The authors also analyze attention patterns in Bagel and proposes a training-free attention adjustment method that biases attention toward task-relevant visual regions identified from context. Empirical results from this intervention show significant improvements on benchmark performance, particularly on rule compliance and visual consistency.

**Compliance With Llm Reviewing Policy:**

Affirmed.

**Final Justification:**

I am maintaining my score of 4 . The benchmark is well-constructed and addresses a genuine gap in the evaluation of context-dependent visual generation; and should be useful to the community.I particularly appreciate the hybrid evaluation protocol with manually authored hints and the diagnostic value of the VQA reformulation experiments.

The rebuttal partially addresses my central concern regarding the grounding of the GFI taxonomy. The authors' argument that IPI/ACE/CKA maps onto established CHC components, can be instantiated as relatively independent generative tasks, and corresponds to diagnosable failure modes is more explicitly articulated than what the paper currently delivers. I note this as a positive signal about the authors' understanding of the issue. However, given claims in the paper such as "we formally define Generative Fluid Intelligence" and "filling a theoretical void to provide foundational guidance," the justification provided does not yet meet the bar for me to increase my score to a 5.

**Key Questions For Authors:**

1. Why is the definition of GFI the authors propose the correct one? Can they better situate their definition in the literature they draw from (Cattel-Horn-Carroll) theory and the wider landscape of proposal for how fluid intelligence should be defined? Are tasks similar to these used to measure fluid intelligence in humans?

2. Could you give a description of what you mean when you say the eval hints are “manually curated”? I assumed this meant human authored but maybe that is not what you meant.

3. The diagnostic story in Section 3.2 felt somewhat unclear to me. Fig. 2(a) seems to suggest that improved context comprehension is an important bottleneck, while Fig. 2(b) argues that the primary issue is execution rather than comprehension. Could the authors clarify their final interpretation of model failure from these results, and whether they view them as pointing mainly to a comprehension gap, an execution gap, or an interaction between the two? Additionally, could the authors comment on why they describe the gains in Fig. 2(a) from different prompting methods as only marginal, particularly for Bagel?

**Limitations:**

No. The paper includes an adequate impact statement, but I did not see a meaningful discussion of the benchmark’s own limitations.In particular, it would help to discuss limitations of the proposed operationalization of GFI, possible sensitivity to the hybrid evaluation protocol and human-curated hints, and the limited scope of the attention-based follow-up analysis.

**Strengths And Weaknesses:**

Strengths:

I like the overall direction of this paper. As image generation quality continues to improve, it feels increasingly important to ask not just whether models can produce convincing-looking images, but whether they can generate images that reflect more flexible, context-dependent reasoning. While some prior benchmarks get at parts of this space, I appreciated the authors’ attempt to make these capabilities more explicit and to evaluate them in a more organized way.

- The benchmark itself appears thoughtfully constructed. The tasks are varied, interesting, and designed with more structure than a lot of broad image-generation evaluations.

- The evaluation setup is fairly rich. Breaking performance into rule compliance, visual consistency, and aesthetic quality gives a more informative view of model behavior than a single overall score, and it is clear the authors put serious effort into making the evaluation workable.

- I found the diagnostic use of the benchmark particularly interesting. The comparison between generation and VQA-style reformulations suggests the benchmark may be useful not only for ranking systems, but also for understanding where failures arise.


Weaknesses:

1. The paper would benefit from more grounding and exposition around its proposed definition of GFI.
    The paper is ambitious in its framing and is effectively arguing for a shift in what kinds of capabilities generative models should be evaluated on. Given that, I think it needs to do more to justify why this particular distillation of GFI is the right one, how it is grounded in the literature it draws from, and how exactly the proposed primitives map onto the benchmark tasks. This is especially true for *Implicit Pattern Induction*, which I found less clearly defined than the other two dimensions. For example, in the “Entity” example in Figure 1, it is not fully clear to me what is meant to be implicit as opposed to simply cross-modal or context-dependent. Because the paper places so much weight on this taxonomy, I think this part of the contribution needs more conceptual grounding and clearer explanation.

2. The positioning relative to prior work, especially in Table 1, feels too categorical.
    The table largely presents prior benchmarks as not evaluating any of the three proposed dimensions, with only a partial exception for UnifyBench. I found this hard to reconcile with the examples in several cited benchmarks. Even if prior work does not use the same terminology or frame these capabilities in the same way, it seems unlikely that none of them test anything resembling explicit constraint execution or contextual adaptation. For example, DPG-Bench includes prompts such as *“a panda bear with aviator glasses on its head”* and *“a black dog sitting between a bush and a pair of green pants standing up with nobody inside them”*. These seem to fit quite naturally under some notion of executing explicit, compositional constraints at generation time. Similarly, benchmarks such as ROVER and RISE appear to include tasks involving pattern completion and logical reasoning that  may overlap at least partially with the paper’s framing (e.g. implicit pattern induction).  While the focus of GENIUS on measuring these dimensions explicitly (i.e. applying a categorization) is front and center, this comparison to previous work muddies my understanding of how they are defining these task dimensions.

3. Some of the paper’s claims feel overstated relative to the evidence presented.
    One is the statement that GENIUS is constructed to “strictly exclude prior knowledge.” In practice, many of the tasks still appear to depend at least partly on prior knowledge of the world or of the concepts being manipulated. For example, the operation-implementation example in Figure 1 seems to require knowing what the operation is in the first place. More broadly, while I agree that these tasks may place greater weight on contextual adaptation and novel reasoning, I do not think the paper convincingly shows that prior knowledge can be cleanly factored out (my read being that fluid intelligence sits on top of crystallized intelligence).
    A second example is the framing that aesthetic fidelity “masks” deep logical deficiencies. I do not think the current evidence supports that strong of a claim; the paper shows coexistence of high visual quality and reasoning failures but its not clear how one masks the other.

4. The attention-adjustment section is interesting, but the mechanistic story feels incomplete.
    I do not view this as a fatal issue for the paper, and the empirical gains on Bagel are clearly interesting. However, I found the method presentation underspecified. In particular, the “semantic relevance map” appears to be central to the method, but I was not able to find a clear definition of how it is actually computed. In addition, I would have liked to see more counterfactual interventions to support the proposed story—for example, using their attention alignment boost on randomly selected visual tokens, or tokens chosen by a simpler heuristic such as highest attention, or some other fixed token-selection pattern. Comparisons of this kind would help establish whether the gains really depend on identifying task-critical cues, rather than on attention reweighting more generally.

5. Figure 2 (a) should not use a stacked bar chart. A grouped bar chart seems much more suitable for showing results across different models. As is it makes it hard to see difference in Nano Banana Pro performance across the conditions. This figure also somewhat contradicts the earlier text that states “empirical results across both Nano Banana Pro and Bagel indicate that these strategies yield only marginal gains.” ([pdf](zotero://open-pdf/library/items/LDT8EUR6?page=5)) , Bagel seems to improve greatly with pre-planning and its possible these improvements would stack.

---

> ### Author Rebuttal · Authors · 2026-03-30
>
> We sincerely thank the **reviewer 2Sop** for recognizing the thoughtful construction and diagnostic richness of GENIUS.
>
> > **W1 & KQ1**: Grounding of GFI & Inductive reasoning.
>
> Our GFI taxonomy is rooted in the CHC theory of cognitive abilities [1,2,3], which decomposes fluid intelligence(FI)  into inductive, deductive, and adaptive reasoning in novel contexts. Our GFI's three dimensions directly map to these established narrow abilities. Specifically, "Implicit Pattern Induction" corresponds to inductive reasoning, a validated transferable component of FI [4]. In our test cases, models must induce rules from both user preferences and dislikes to reflect them in the final image.
>
> > **W2**: Prior work comparison & task definition.
>
> We acknowledge that the potential categorical categorization in Table 1 and will conduct a comprehensive manual re-audit of cited benchmarks for the final version. While DPG-Bench assesses the compositional capability, GENIUS emphasizes ad-hoc binding, requiring models to interpret and visualize non-semantic symbols (e.g., `<WN>`) based purely on immediate context. Similarly, our "Implicit Pattern Induction" targets discovering visual patterns within user intent rather than simple pattern completion. For instance, in the "Entity" task, the model must extract unnamed aesthetic styles or specific character traits from provided examples.
>
> > **W3**: Prior Knowledge & Aesthetic Masking.
>
> Regarding prior knowledge, we agree that total isolation of foundational visual common sense (necessary for basic image formation) is impractical. Our core objective is isolating the external world prior to test fluid adaptability, mimicking how a human can analogize and draw a novel creature based purely on three local examples. We will refine our claim to "isolating specific external world priors" while acknowledging the role of basic visual priors. "Aesthetic masking" reflects our insight into a community-wide inductive bias: an overemphasis on image quality has allowed high visual fidelity to mask deep logical deficiencies, creating an illusion of capability.
>
> > **W4**: Computation of the Semantic Relevance Map & Counterfactual Interventions
>
> We extract keywords $K_i$ for each context image $I_i$. $S_i$ ($S$ is the concated result of  each $S_i$) is computed by calculating the similarity between $I_i$'s semantic embeddings and $K_i$'s token embeddings. This yields a spatial probability map that is resized to the VAE embedding dimensions. We will add the detailed mathematical formulation in the revision. We tested applying attention adjustment to randomly selected visual tokens. As the results below show, random reweighting fails to improve performance (Check Bagel(-ours)'s performance in the main results for comparison). This confirms that our gains rely strictly on emphasizing task-critical cues, rather than generic attention manipulation.
>
> |Method|Overall|IP-RC|IP-VC|IP-AQ|SC-RC|SC-VC| SC-AQ| VC-RC | VC-VC | VC-AQ | PC-RC | PC-VC | PC-AQ | MS-RC | MS-VC | MS-AQ |
> |---|:-:|:-:|:-:|:-:|:-:|:-:|:-:|:-:|:-:|:-:|:-:|:-:|:-:|:-:|:-:|:-:|
> |Bagel-Random|26.15|26.12| 26.45|83.74|29.03|15.41 | 75.88 | 21.77 | 12.50 |48.51|21.69|16.57|73.91 | 32.93 |-|52.94|
>
> > **W5 & KQ3**: Visualization & Benchmark results.
>
> We will update Fig 2(a) to a grouped bar chart in the final version as suggested. We emphasize that the results are complementary rather than contradictory, revealing a complex interaction between understanding and execution. While prompting improves understanding, gains are capped by the "execution gap"; conversely, generative performance is bounded by prompts. This dual bottleneck proves GFI can not be resolved by isolating either module. Our description of gains as "marginal" is relative to other benchmarks: Bagel+CoT yields a ~20-point gain on WISE but only ~10 on GENIUS, highlighting the difficulty of GFI. Regarding the potential stacking of improvements, we tested combining pre-planning and post-reflection. As shown in the results below, the slight overall score increase falls far short of a synergistic effect and does not justify the significantly higher token consumption.
>
> |Method|Original|+ Pre-planing|+ Post-reflection|+ All|
> |---|:-:|:-:|:-:|:-:|
> |Bagel|26.74|30.40|34.19|35.15|
> |Nano Banana Pro|57.19|67.39|70.47|72.59|
>
> > **KQ2**: Regarding "Manually Curated" Hints.
>
> We confirm that all Eval-Hints are human-authored (hand-written) by the authors. This reflects the significant manual effort we invested to ensure the rigor and quality of our evaluation protocol.
>
> We thank the reviewer again for highlighting these points. We will incorporate the suggested clarifications, expand the discussion, and strengthen comparisons in the final version.
>
> [1] Human Cognitive Capabilities: Gf-Gc Theory
>
> [2] The Cattell–Horn–Carroll Theory of Cognitive Abilities
>
> [3] Human Cognitive Abilities: A Survey of Factor-Analytic Studies
>
> [4] Inducing inductive reasoning: Does it transfer to fluid intelligence?

---

> > ### Author Rebuttal · Reviewer_2Sop · 2026-04-03
> >
> > Thank you for the detailed rebuttal. I found the clarifications helpful, especially on the human-authored eval hints, the softened prior-knowledge claim, and the additional results on random-token reweighting, which made the Bagel follow-up more convincing to me.
> >
> > My main remaining reservation is around the grounding of the proposed GFI taxonomy. While the rebuttal better situates the three dimensions in CHC theory, I still do not feel the paper fully justifies why this particular decomposition is the right operationalization of generative fluid intelligence, as opposed to one plausible decomposition among several. The additional citations are helpful, but I do not think citation alone fully resolves this point; I would still welcome more elaboration on why the authors view this as the appropriate decomposition for multimodal generation specifically, and how they see it as positioned relative to plausible alternatives. Because the paper leans so heavily on this framing, I think it would benefit from stronger grounding and motivation than it currently has.
> >
> > Overall, the rebuttal improves my understanding of the paper and resolves several concrete questions. It strengthens my confidence in the paper, and I remain positive on it.

---

> > > ### Author Response · Authors · 2026-04-06
> > >
> > > Thank you for the thoughtful follow-up. We do not claim that this is the only possible taxonomy; however, for the specific setting of multimodal generation, we believe it is **the most natural and most appropriate operational decomposition**. This is because it satisfies three criteria at once: first, it maps directly onto induction, deduction, and adaptation as core components of fluid intelligence; second, the three dimensions can be clearly instantiated as relatively independent generative tasks; and third, they correspond to observable and diagnosable failure modes in current models. In this sense, the decomposition is not only theoretically motivated, but also empirically discriminative.
> > >
> > > Our central view is that GFI is not generic reasoning, but dynamic generation driven by immediate context. Concretely, **Implicit Pattern Induction (IPI)** induces “soft” information from multimodal context to uncover latent contextual structure; **Ad-hoc Constraint Execution (ACE)** reflects newly defined “hard” logic in the generated output; and **Contextual Knowledge Adaptation (CKA)** filters interference by suppressing outdated pretrained biases and resolving conflicts between prior knowledge and current context. In short, the three dimensions correspond to discovering new structure, executing new rules, and inhibiting stale priors, which together capture the core context-driven abilities required for multimodal generation.
> > >
> > > We believe this decomposition is especially suitable for generation tasks, because generation requires not only understanding the context, but also faithfully realizing that understanding in the visual output.
> > >
> > > We also acknowledge that alternative perspectives such as planning, memory, and compositionality are possible. However, in multimodal generation, these higher-level descriptions still tend to reduce to three more basic questions: what new structure must be extracted from context, how it should be turned into generation rules, and how the model should adapt when those rules conflict with pretrained priors. Consider a compositional generation example with long context and complex novel concepts: the model must first infer the user’s preferred character style or the target relations in the image from context, which corresponds to IPI; it must then respond to newly introduced generation constraints in the context and integrate new concepts into the output, which corresponds to ACE; and if the context conflicts with the model’s default common sense or pretrained preferences, such as physical regularities, it must suppress those priors and follow the current context, which corresponds to CKA. Therefore, in the final version, we will state more explicitly that IPI/ACE/CKA is not the only theoretical decomposition of GFI, but it is, in our view, the most appropriate, actionable, and diagnostically useful one for multimodal generation.

---

### Official Review · Reviewer_c6T6 · 2026-03-17

**Soundness:** 3
**Presentation:** 2
**Significance:** 2
**Originality:** 2
**Overall Recommendation:** 4
**Confidence:** 3

**Summary:**

This paper introduces GENIUS, a benchmark for evaluating Generative Fluid Intelligence (GFI) in unified multimodal image generation. The authors argue that existing benchmarks largely test Crystallized Intelligence, which relies on recalling accumulated knowledge and learned schemas, while under-testing the ability to induce patterns, reason through constraints, and adapt to novel scenarios on the fly. To address this, they define GFI along three dimensions: Implicit Pattern Induction, Ad-hoc Constraint Execution, and Contextual Knowledge Adaptation. They instantiate these ideas in a benchmark of 510 expert-curated samples spanning 5 tasks and 20 sub-tasks. Evaluation is based on three metrics: Rule Compliance, Visual Consistency, and Aesthetic Quality, using manually curated hints together with an LMM-based judge.

The empirical study evaluates 12 representative models and finds substantial headroom: both proprietary system and open-source VLM perform limited. The paper further argues that the largest weakness appears in contextual adaptation, where models tend to default to pretrained priors rather than follow the local multimodal context. To support the evaluation protocol, the authors report strong agreement between two LMM judge and human experts on a 100-sample study. Finally, motivated by attention analyses on Bagel, the paper proposes a training-free attention adjustment mechanism and reports consistent gains on Bagel.

**Compliance With Llm Reviewing Policy:**

Affirmed.

**Final Justification:**

The rebuttal is helpful and addresses my concerns substantively. I decide to keep my positive score.

**Key Questions For Authors:**

1. How did authors ensure that the 510 benchmark instances adequately cover the intended concept space, and can you provide any stability analysis over benchmark subsets or alternative task mixtures?
2. The paper validates LMM-as-a-judge on 100 samples from two representative models. Can you expand on whether the same level of agreement holds across a broader set of models and task types?

**Limitations:**

No limitation discussion is provided. I think the limitations discussion should be more explicit about benchmark-scale constraints, construction bias, judge dependence, and cross-interface comparability.

**Strengths And Weaknesses:**

Strength:
1. This paper identifies a real gap in current evaluation on text-to-image models. The benchmark is designed to probe context-bound, rule-sensitive generation where the correct output depends on integrating interleaved multimodal context rather than recalling familiar concepts.
2. The benchmark design is fairly thoughtful. The paper covers three clearly differentiated dimensions, five concrete tasks, and twenty sub-tasks, and uses a hybrid evaluation protocol. I also appreciate the use of manually annotated hints and the anti-plagiarism check in the visual-consistency metric.
3. The paper makes a credible effort to validate its evaluation protocol. The LMM-as-a-judge setup is compared against human ratings on 100 samples from two representative models.

Weakness:
1. The conceptual framing around “fluid intelligence” feels broader than what is actually established. The benchmark clearly measures useful aspects of context-following and rule adaptation, but the paper sometimes presents this as a more general statement about intelligence than the experiments fully justify. I would prefer a more conservative framing of what the benchmark does and does not measure.
2. The benchmark scale is still modest relative to the scope of the claims. While 510 manually curated examples is understandable given the annotation burden, this is still a limited sample size for operationalizing such a broad concept, and it leaves open questions about coverage and ranking stability.
3. The evaluation remains substantially dependent on automated judging. The human-validation study is helpful, but it covers only 100 outputs from two models. I would be more convinced by a broader manual audit across more systems and task types.
4. Cross-model comparison is not perfectly clean because of input-format differences. The main experiments use interleaved inputs for some models and decoupled/edit-style inputs for others, and the appendix shows that performance can vary significantly with formatting. This makes it harder to interpret the leaderboard as a pure capability ranking.
5. The mechanistic explanation is still somewhat preliminary. The attention-based diagnosis is interesting, but the causal claim that attention defects are the primary source of failure is only partially supported, and the proposed intervention is evaluated only on Bagel, so it currently reads more like a promising case study than a general explanation.

---

> ### Author Rebuttal · Authors · 2026-03-30
>
> We sincerely thank the **reviewer c6T6** for the effort in reviewing our paper. Our responses according to the reviewer's comments are summarized as follows.
>
> > **W1**: The conceptual framing around “fluid intelligence” feels broader than what is actually established.
>
> We acknowledge that "Fluid Intelligence" (FI) is a broad concept; thus, we strictly constrain our framework to the context of visual generation, specifically in Line 020-021. Representative FI benchmarks ARC-AGI, focus on inducing novel patterns from a given context to solve problems, a capability directly encapsulated within our "Ad-hoc Constraint Execution" dimension. By extending the core evaluating dimension of ARC-AGI into generation domain, GENIUS effectively broadens the scope of FI evaluation, justifying its classification as a specialized form of FI. While human fluid intelligence is multifaceted, we systematically selected three highly representative dimensions, providing a sufficiently rigorous and targeted assessment within the defined scope of Generative FI.
>
> > **W2 & KQ1**: The benchmark scale, coverage and its evaluating robustness.
>
> For evaluating UMM, the quality of samples are more critical than sheer quantity. Unlike benchmarks relying on large-scale synthetic data, all 510 samples in GENIUS are expert-curated. The cost of collecting high-quality interleaved multimodal data and expert annotation significantly exceeds that of simple text prompts. For comparison, the high-difficulty benchmark RISE[1] (NIPS 2025 oral) contains only 360 samples. We meticulously ensured sample diversity, including various scenes, demographics (gender/age) and so on.  Furthermore, relative rankings remain highly consistent across different judges (Gemini vs. Qwen). To further demonstrate ranking stability, we newly add 90 curated samples uniformly coverage all tasks and evaluate representative models (Due to the space limit). The results show the ranking remains consistent, demonstrating the current subset is sufficient for distinguishing models:
>
> | Method | Overall | IP-RC | IP-VC | IP-AQ | SC-RC | SC-VC | SC-AQ | VC-RC | VC-VC | VC-AQ | PC-RC | PC-VC | PC-AQ | MS-RC | MS-VC | MS-AQ |
> |---|:---:|:---:|:---:|:---:|:---:|:---:|:---:|:---:|:---:|:---:|:---:|:---:|:---:|:---:|:---:|:---:|
> | Nano Banana Pro | 55.64 | 65.12 | 42.28 | 95.83 | 69.42 | 47.23 | 91.59 | 75.32 | 65.79 | 97.08 | 50.73 | 39.81 | 89.66 | 32.82 | - | 93.88 |
> | GPT-Image | 44.56 | 56.08 | 40.44 | 92.83 | 56.51 | 30.87 | 93.13 | 47.33 | 63.02 | 91.56 | 41.09 | 32.07 | 89.62 | 25.91 | - | 83.88 |
> | Qwen-Image | 29.55 | 34.26 | 25.25 | 69.92 | 33.91 | 25.81 | 70.31 | 25.11 | 43.80 | 56.19 | 25.23 | 18.98 | 70.82 | 23.26 | - | 68.33 |
> | Bagel | 24.56 | 24.39 | 25.16 | 83.48 | 26.87 | 14.57 | 75.41 | 20.52 | 11.73 | 46.96 | 20.64 | 17.67 | 74.06 | 30.91 | - | 52.32 |
>
> > **W3 & KQ2**: The evaluation process and human study.
>
> GENIUS employs a hybrid protocol grounding the LMM judge in expert-curated "Eval-Hints" (gold standards). We expanded human validation to all 510 samples, maintaining a Pearson correlation which is 0.972. This confirms our framework as a robust and reliable alternative to manual evaluation. The full audit results and expanded correlation plots will be integrated into the final version.
>
> > **W4**: Cross-model comparison
>
> The format disparity was not an intentional design choice but a reflection of the fact that most current models do not support true interleaved inputs. This sensitivity itself highlights a key limitation in the fluid adaptability of existing UMMs, underscoring the diagnostic value of GENIUS. To ensure a clearer and fairer comparison, we will provide a capability ranking categorized by input format (e.g., interleaved vs. decoupled) in the final version.
>
> > **W5**: The mechanistic explanation is still somewhat preliminary.
>
> Our analysis explicitly provides a "possible view" (Line 103～104), leaving other potential failure modes for future community research. To further validate generalizability, we implemented the intervention strategy on GLM-Image[2]: consistent qualitative (https://anonymous.4open.science/r/genius_rebuttal-74BB/rebuttal.png) and quantitative  improvements demonstrate its cross-architecture effectiveness and robustness.
>
> | Method | Overall | IP-RC | IP-VC | IP-AQ | SC-RC | SC-VC | SC-AQ | VC-RC | VC-VC | VC-AQ | PC-RC | PC-VC | PC-AQ | MS-RC | MS-VC | MS-AQ |
> |---|:---:|:---:|:---:|:---:|:---:|:---:|:---:|:---:|:---:|:---:|:---:|:---:|:---:|:---:|:---:|:---:|
> | GLM-Image | 24.71 | 32.94 | 19.86 | 93.53 | 22.37 | 21.15 | 87.50 | 27.50 | 12.50 | 70.83 | 20.30 | 15.52 | 71.29 | 17.73 | - | 70.91 |
> | GLM-Image-Ours | 30.04 | 39.87 | 37.18 | 82.78 | 26.01 | 27.44 | 81.23 | 30.67 | 32.50 | 60.33 | 20.89 | 33.27 | 63.23 | 17.80 | - | 65.62 |
>
> [1] Envisioning Beyond the Pixels: Benchmarking Reasoning-Informed Visual Editing
>
> [2] GLM-Image: Auto-regressive for Dense-knowledge and High-fidelity Image Generation

---

> > ### Author Rebuttal · Reviewer_c6T6 · 2026-04-02
> >
> > The rebuttal is helpful and addresses several of my concerns in a substantive way. In particular, I appreciate the added evidence on ranking stability, the expansion of human validation to the full benchmark, and the additional cross-model evidence for the proposed intervention. These additions make the evaluation protocol and diagnostic analysis more convincing. I still think the framing around “Generative Fluid Intelligence” is somewhat broader than what is fully established, and the benchmark remains modest in scale relative to the ambition of the concept. In addition, cross-format comparability should be presented more explicitly in the final version. That said, the rebuttal overall strengthens my confidence in the paper’s contribution, and my overall assessment remains positive.

---

> > > ### Author Response · Authors · 2026-04-06
> > >
> > > Thank you very much for taking the time to read our rebuttal and for your positive follow-up comments. We sincerely appreciate your thoughtful feedback and are glad that our clarifications helped address your concerns. Your comments have been very helpful in improving the presentation and rigor of the paper. We will carefully incorporate the rebuttal clarifications and new experimental results into the final version.

---

### Decision · Program_Chairs · 2026-04-30

**Decision:**

Reject

**Comment:**

The GENIUS benchmark addresses a critical gap in multimodal evaluation by shifting from static knowledge recall to "Generative Fluid Intelligence (GFI)", the capacity of models to reason through novel, context-bound constraints. The work is technically solid, featuring an expert-curated dataset and a robust evaluation protocol that utilizes human-authored "Eval-Hints" to ground automated judging. During the rebuttal, the authors substantively strengthened the submission by demonstrating ranking stability, expanding human validation, and proving that their attention-intervention strategy generalizes across different model architectures.

Despite these strengths, the term "Generative Fluid Intelligence" remains a conceptual overclaim, as it is a broad psychological construct that the paper operationalizes through a specific, albeit plausible, three-part taxonomy. While the authors defended this framing as a natural decomposition for generative tasks, the AC strongly suggests that the authors revise the manuscript to position their definitions as one actionable framework rather than a definitive definition.